# Columnar processing of border ownership in primate visual cortex

**Tom P Franken\*, John H Reynolds**

Systems Neurobiology Laboratory, The Salk Institute for Biological Studies, La Jolla, United States

**Abstract** To understand a visual scene, the brain segregates figures from background by assigning borders to foreground objects. Neurons in primate visual cortex encode which object owns a border (border ownership), but the underlying circuitry is not understood. Here, we used multielectrode probes to record from border ownership-selective units in different layers in macaque visual area V4 to study the laminar organization and timing of border ownership selectivity. We find that border ownership selectivity occurs first in deep layer units, in contrast to spike latency for small stimuli in the classical receptive field. Units on the same penetration typically share the preferred side of border ownership, also across layers, similar to orientation preference. Units are often border ownership-selective for a range of border orientations, where the preferred sides of border ownership are systematically organized in visual space. Together our data reveal a columnar organization of border ownership in V4 where the earliest border ownership signals are not simply inherited from upstream areas, but computed by neurons in deep layers, and may thus be part of signals fed back to upstream cortical areas or the oculomotor system early after stimulus onset. The finding that preferred border ownership is clustered and can cover a wide range of spatially contiguous locations suggests that the asymmetric context integrated by these neurons is provided in a systematically clustered manner, possibly through corticocortical feedback and horizontal connections.

**\*For correspondence:**
tfranken@salk.edu

**Competing interest:** The authors declare that no competing interests exist.

## Editor's evaluation

Franken and Reynolds use multielectrode laminar recordings in V4 of macaques to study whether border ownership (BO) occurs with varying latency across layers, whether their ownership preference is consistent across layers, and the relationship between BO and orientation preference. They convincingly show that BO emerges in deep layers and is consistent across depth. Based on these findings, they suggest that BO could arise from horizontal and feedback connections rather than feedforward connections.

## Introduction

One of the deepest and most enduring mysteries of visual perception is how the brain constructs an internal model of the ever-changing world that falls before our eyes. An essential step in this dynamic process of constructive perception is the assignment of border ownership. Consider panel 1 in *Figure 1A*. The dashed circle indicates the classical receptive field (cRF) of a hypothetical neuron, straddling the edge of an object. The portion of the edge falling inside the circle is perceived to be owned by the light gray square on the bottom left of the edge. Now consider panel 2 in *Figure 1A*. The local edge within the circle is identical to that in the left panel but this local edge is now perceived as owned by the dark gray square to the upper right of the edge. This phenomenon is called border ownership. It was first recognized by the Gestalt psychologists in the early 20th century and is beautifully illustrated in Rubin's famous face-vase illusion (*Koffka, 1935*; *Rubin, 1921*). Border ownership

**eLife digest** To understand a visual scene, the brain needs to identify objects and distinguish them from background. A border marks the transition from object to background, but to differentiate which side of the border belongs to the object and which to background, the brain must integrate information across space. An early signature of this computation is that brain cells signal which side of a border is 'owned' by an object, also known as border ownership. But how the brain computes border ownership remains unknown.

The optic nerve is a cable-like group of nerve cells that transmits information from the eye to the brain's visual processing areas and into the visual cortex. This flow of information is often described as traveling in a feedforward direction, away from the eyes to progressively more specialized areas in the visual cortex. However, there are also numerous feedback connections in the brain, running backward from more specialized to less specialized cortical areas.

To better understand the role of these feedforward and feedback circuits in the visual processing of object borders, Franken and Reynolds made use of their stereotyped projection patterns across the cortex layers. Feedforward connections terminate in the middle layers of a cortical area, whereas feedback connections terminate in upper and lower layers. Since time is required for information to traverse the cortical layers, dissecting the timing of border ownership signals may reveal if border ownership is computed in a feedforward or feedback manner. To find out more, electrodes were used to record neural activity in the upper, middle and lower layers of the visual cortex of two rhesus monkeys as they were presented with a set of abstract scenes composed of simple shapes on a background.

This revealed that cells signaling border ownership in deep layers of the cortex did so before the signals appeared in the middle layer. This suggests that feedback rather than feedforward is required to compute border ownership. Moreover, Franken and Reynolds found evidence that cells that prefer the same side of border ownership are clustered in columns, showing how these neural circuits are organized within the visual cortex.

In summary, Franken and Reynolds found that the circuits of the primate brain that compute border ownership occur as columns, in which cells in deep layers signal border ownership first, suggesting that border ownership relies on feedback from more specialized areas. A better understanding of how feedback in the brain works to process visual information helps us appreciate what happens when these systems are impaired.

---

represents a fundamental computation in visual perception that is thought to be critical to visual scene segmentation and object recognition (*Nakayama et al., 1995*). Von der Heydt et al. discovered neurons in primate visual cortex that are selective for border ownership, most prominently in extrastriate visual areas V2 and V4 (*Zhou et al., 2000*).

Though the existence of border ownership-selective neurons has been well established in prior studies using single electrodes (*Hesse and Tsao, 2016*; *O'Herron and von der Heydt, 2009*; *Zhang and von der Heydt, 2010*), how this selectivity arises from cortical circuits remains unclear (*Grossberg, 2015*; *von der Heydt, 2015*; *Wagatsuma and Sakai, 2016*; *Yazdanbakhsh and Livingstone, 2006*). Some authors have proposed a dominant role for feedforward inputs, which carry information from upstream areas (*Sakai et al., 2012*; *Sakai and Nishimura, 2006*; *Supèr et al., 2010*), whereas others posit a central role for horizontal connections and/or cortical feedback (*Craft et al., 2007*; *Grossberg, 2015*; *Hu and Niebur, 2017*; *Zhang and von der Heydt, 2010*; *Zhaoping, 2005*). These pathways have a distinct laminar organization: feedforward inputs arrive primarily in the granular (input) layer, whereas horizontal and corticocortical feedback connections predominantly target superficial and deep layers (*Douglas and Martin, 2004*; *Rockland and Pandya, 1979*; *Rockland et al., 1994*; *Rockland and Lund, 1983*; *Ungerleider and Desimone, 1986*; *Yoshioka et al., 1992*). The laminar timing of border ownership may thus give clues regarding the roles of these pathways in this computation. Here, we used linear multielectrode probes to compare onset times of border ownership selectivity between laminar compartments.

It is also unknown how the preferences of border ownership-selective neurons in different layers relate to each other. This is in contrast to orientation preference, which is well known to be spatially

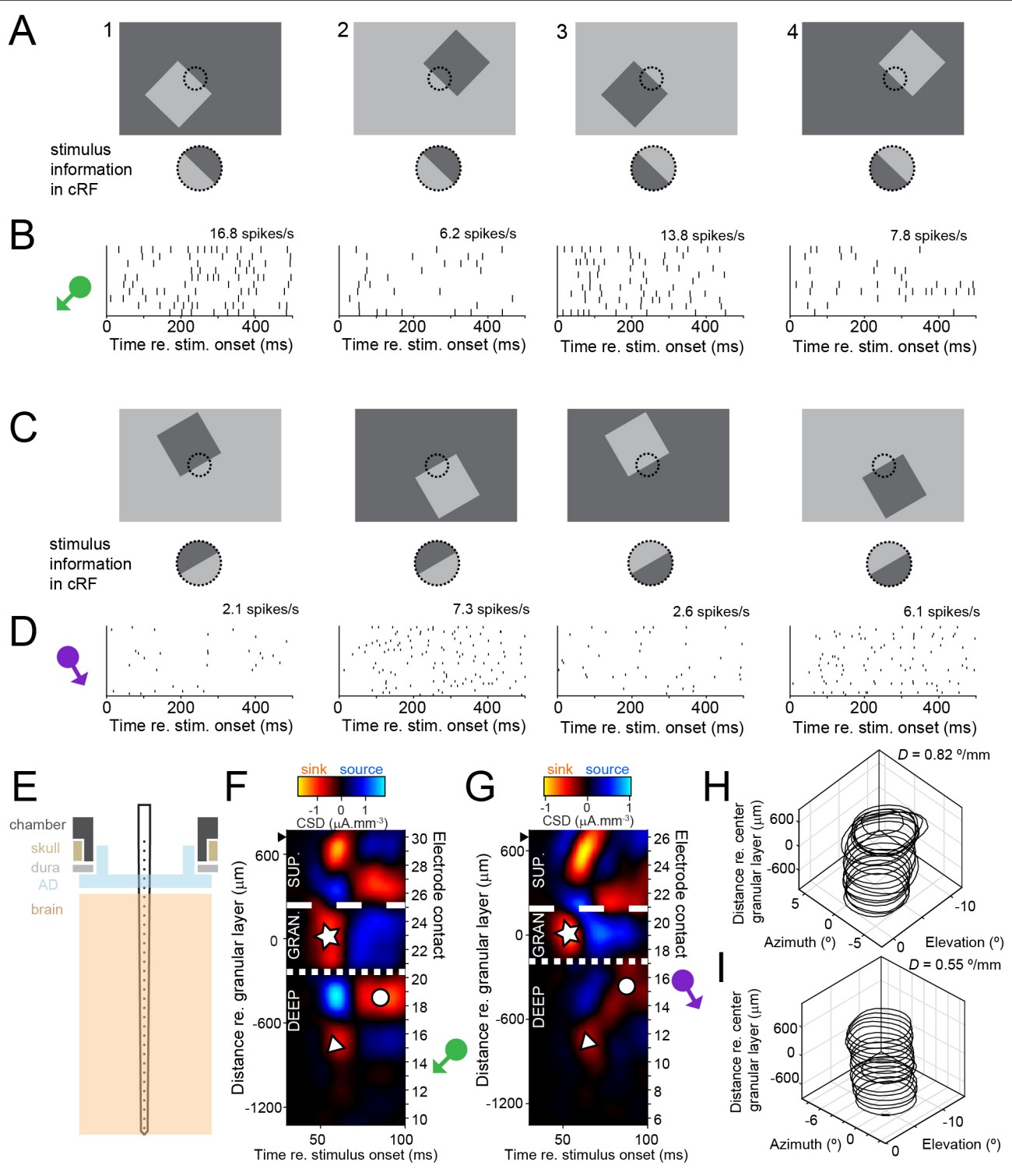

**Figure 1.** Laminar multielectrode recordings from border ownership-selective units in area V4. (**A**) Top row shows set of border ownership stimuli. Black dotted outline represents the classical receptive field (cRF). Bottom row shows that the stimulus information in the cRF is identical for stimuli 1 and 2, and for stimuli 3 and 4. (**B**) Dot rasters showing responses to the stimuli in A from a border ownership-selective well-isolated unit. The symbol on the left indicates the preferred side of border ownership for the unit. Average spike rates in the stimulus window are indicated above the panels. (**C, D**) Similar to A and B, for a multiunit cluster recorded during a different penetration. (**E**) Cartoon showing the recording setup. A laminar multielectrode probe with 32 channels was lowered through a transparent artificial dura (AD), orthogonal relative to the cortical surface. (**F**) Laminar compartments (superficial; granular [input]; deep layers) were estimated using current source density (CSD) analysis. See text for definitions of the compartments and

*Figure 1 continued on next page*

*Figure 1 continued*

explanation of symbols. Distance from center of granular layer is shown along ordinate on the left, and the number of electrode contact on the right (higher numbers correspond to more superficial contacts). Black solid triangle indicates position of most superficial electrode contact on which multiunit activity was recorded. Data are from the same penetration during which the unit in B was recorded. The position of the green symbol indicates that this unit was positioned in the deep layers. See also *Figure 1—figure supplements 1–3*. (**G**). Similar to F, for the penetration during which the unit shown in D was recorded. (**H**) Receptive field contours for multiunit activity recorded on different electrode contacts from the penetration shown in F. Contours are drawn at $z = 3$. Orthogonality of the penetration was evaluated by fitting a line through the centers of the receptive field contours and computing $D$, the distance between pairs of azimuth and elevation coordinates of receptive field centers per mm depth (Methods). (**I**) Similar to H, for the penetration shown in G.

The online version of this article includes the following figure supplement(s) for figure 1:

**Figure supplement 1.** Construction of current source density (CSD) map.

**Figure supplement 2.** Additional examples of current source density (CSD) maps.

**Figure supplement 3.** Consistency of current source density (CSD) maps between studies and cortical areas.

organized in the primate brain. Primate areas V1 and V2 contain orientation columns (*Hubel and Livingstone, 1987*; *Hubel et al., 1978*; *Vanduffel et al., 2002*). In V4, imaging studies indicate that there is at least clustering of orientation preference in superficial layers of V4 (*Ghose and Ts'o, 1997*; *Roe et al., 2012*; *Tanigawa et al., 2010*), although it is unclear whether these clusters are columnar. Nor do we know if border ownership preference is organized in columns. The border ownership of a given border is defined entirely by asymmetries outside the cRF (as opposed to the border's orientation). A systematic organization of border ownership preference would therefore imply a clustered arrangement of the neural pathways that underlie these asymmetries.

Finally, the relation between border ownership selectivity and orientation tuning is unclear. Prior studies have focused on orientation-selective units and tested border ownership at the neuron's preferred orientation, without examining the relationship between orientation preference and border ownership preference (*Hesse and Tsao, 2016*; *Zhang and von der Heydt, 2010*; *Zhou et al., 2000*). One possibility is that border ownership is fundamentally a border property, such that border ownership selectivity is maximal for the preferred border orientation. Another possibility is that border ownership selectivity rather represents a surface signal (*Grossberg, 2015*), and may thus be less strictly tied to border orientation and orientation tuning.

Here, we addressed these questions using laminar multielectrode probes to record from border ownership-selective units in macaque area V4 across layers. We replaced the native dura with a transparent artificial dura (AD) to enable us to reliably position the probe normal to the cortical surface, on the relatively narrow exposed surface of area V4. We compared the timing and magnitude of border ownership selectivity across laminar compartments. If border ownership selectivity in V4 is inherited from V2, its dominant source of cortical input (*Markov et al., 2011*), we expect to see it early and prominently in neurons in the granular layer, which is the main target of this projection (*Rockland, 1992*; *Gattass et al., 1997*). Next, we compared the functional organization of border ownership preference across layers, to test whether the preferred side of ownership is shared. Finally, we examined the relationship between border ownership preference and border orientation preference.

## Results

To elucidate the functional organization and timing of border ownership we used laminar recording electrodes oriented normal to the cortical surface to record from well-isolated units and multiunit activity in macaque area V4, during fixation. We present data from two rhesus macaques recorded over 88 penetrations (animal Z: 34; animal D: 54). Laminar analyses are performed for 81 penetrations for which the current source density (CSD) analysis (see below and Methods) was interpretable. For each penetration, we first obtained data to map the receptive fields (RFs) by presenting fast sequences of Gaussian windowed luminance contrast edges and rings at random positions in the appropriate visual quadrant. The orthogonal position of the probe relative to the cortical surface resulted in a vertical stacking of the RFs from different electrode contacts (*Figure 1H,I*; *Figure 1—figure supplement 1C*). Orthogonality was evaluated by computing the distance between RF centers per mm depth along the probe ($D$ in *Figure 1H,I*; *Figure 1—figure supplement 1C*; Methods; median [interquartile range] 0.83°/mm [1.00°/mm], $n = 81$ penetrations). At this retinotopic location we then obtained data

to determine orientation tuning (using luminance contrast edges of different orientations) and to compute CSD (see below). We then presented the border ownership stimuli (*Figure 1A and C*). These stimuli were similar to those that have previously been used to measure border ownership tuning (*Zhou et al., 2000*). We carefully positioned square stimuli such that one edge (termed the 'central edge') fell within the cRF, while ensuring that the stimulus features that defined border ownership (the other three edges and the four corners of the square) all fell outside of the cRF. Units for which the stimulus was not properly placed relative to the cRF were not included in the analysis (see inclusion criteria in Methods). This resulted in 685 well-isolated units (animal Z: 227; animal D: 458) and 765 multiunit clusters (animal Z: 329; animal D: 436).

*Figure 1B* shows the responses from a well-isolated unit to the stimuli in *Figure 1A* as raster plots. The unit fires more to stimulus 1 than to stimulus 2, even though the stimulus information in the cRF is identical in both cases. In other words, this unit fires more to an identical contrast edge in the cRF if that edge is owned by an object on the lower left compared to when it is owned by an object on the upper right. To test whether the difference in spike rates could be explained by the difference in luminance of the square object between stimuli 1 and 2, we also reversed all luminances (resulting in stimuli 3 and 4 in *Figure 1A*). Again we observe that this unit prefers the stimulus where the central edge is owned by an object on the lower left (prefers stimulus 3 over stimulus 4, *Figure 1B*). Together, we conclude that this unit's preferred side of border ownership is the lower left side of the border, indicated by the direction of the green arrow in *Figure 1B*. We assessed statistical significance of border ownership selectivity with a permutation test on the absolute value of the border ownership index ($BOI$, see Methods, $|BOI| = 0.38$ for this unit, permutation test $p < 0.0001$).

To estimate the laminar position of units, we performed current source density (CSD) analysis of the local field potential (LFP) evoked by small rings in the cRF (Methods; *Figure 1—figure supplement 1*; *Nandy et al., 2017*; *Mitzdorf, 1985*; *Poort et al., 2016*; *Bijanzadeh et al., 2018*; *Ferro et al., 2021*). This analysis results in a pattern of current sinks and current sources (*Figure 1F* shows an example CSD map for the penetration during which the unit in *Figure 1B* was recorded). Such maps show a prominent leading current sink in the central portion of the penetration (white star in *Figure 1F*), with a current source followed by a current sink on the electrode contacts positioned deep from it (white disc in *Figure 1F*). Below this current source, we typically observe a current sink with longer latency (white triangle in *Figure 1F*). This sink–source pattern occurred consistently in different penetrations (*Figure 1G*; *Figure 1—figure supplement 1*; *Figure 1—figure supplement 2*). This consistency between penetrations resulted in an average CSD map across penetrations that was very similar to that of individual penetrations (*Figure 1—figure supplement 3A*). Studies from other laboratories in behaving macaques have described this sink–source pattern as well in area V4 (*Pettine et al., 2019*, their Figure 1C; *Lu et al., 2018*, their Figure S4B) but also downstream in the medial temporal cortex (*Takeuchi et al., 2011*, area 36, their Figure S1, note that the ordinate is reversed in this figure relative to ours). These sink–source patterns are very similar to those in our data (compare panels A–C in *Figure 1—figure supplement 3*). Takeuchi et al. paired this analysis with histological verification by applying electrolytic marks during the recordings and found that the prominent early current sink indicates the position of the granular layer (*Figure 1—figure supplement 3C*), thus consistent with CSD analysis in V1 (*Mitzdorf, 1985*). The very similar sink–source patterns between individual penetrations, and between studies from different laboratories, including work from others that has been validated histologically, gives us confidence that we can meaningfully use these patterns to assign electrode contacts to granular, superficial, and deep layers, as other laboratories have done (e.g., *Lu et al., 2018*; *Ferro et al., 2021*). We therefore draw boundaries above and below this current sink and identify this compartment as the granular layer (white star, between white dashed and dotted lines in *Figure 1F*). From the position of the electrode contact on which each unit was recorded relative to the CSD map, we could then locate units in superficial, granular, or deep layer compartments (see Methods for classification criteria). For example, the unit shown in *Figure 1B* was located in the deep compartment. *Figure 1C,D,G,I* show another example (multiunit cluster), recorded in deep cortical layers from a different penetration ($|BOI| = 0.51$, $p < 0.0001$).

Across the population, we find border ownership selectivity in 51.1 % of well-isolated units (350 out of 685 units; pooling across well-isolated and multiunit clusters, we find border ownership selectivity in 44.6 % of units [647 out of 1450 units]). This proportion is high in all laminar compartments (superficial: 43.3 % [58 out of 134]; granular: 57.3 % [102 out of 178]; deep: 56.0 % [121 out of 216]; pooled

well-isolated and multiunit clusters: superficial: 42.8 % [140 out of 327]; granular: 51.5 % [167 out of 324]; deep: 49.1 % [210 out of 428]).

## Border ownership selectivity occurs first in deep cortical layers

*Figure 2A* shows the time course of the responses evoked by the preferred (solid red line) and non-preferred (dashed blue line) side of border ownership, averaged over all well-isolated units that are selective for border ownership. Consistent with prior studies (*Zhou et al., 2000*), we observe that border ownership selectivity (difference in response to the preferred and the non-preferred side) occurs early after onset of the stimulus-evoked response. The asterisk indicates when the difference between these functions first becomes statistically significant (56.5 ms after stimulus onset; sign rank test p < 0.05 for 20 adjacent ms). When evaluating these functions in each laminar compartment we observe that these functions diverge substantially early after onset in deep cortical layers (*Figure 2D*; significant at 58.8 ms), but later in superficial (*Figure 2B*; significant at 85.8 ms) and granular layers (*Figure 2C*; significant at 67.3 ms). This definition of latency depends on sample size, but a subsampling analysis shows that differences in sample size between layers do not explain the shorter latency for deep layers (*Figure 2—figure supplement 1*). To compare the time course of border ownership modulation between layers, we defined the BOI function *B* for each laminar compartment as

$$B\left(t\right) = \frac{R_{\mathrm{pref}}\left(t\right) - R_{\mathrm{non-pref}}\left(t\right)}{R_{\mathrm{pref}}\left(t\right) + R_{\mathrm{non-pref}}\left(t\right)}$$

where $R_{pref}(t)$ and $R_{non-pref}(t)$ are, respectively, the evoked response functions to the preferred and the non-preferred sides of border ownership (red and blue dashed functions plotted in *Figure 2A–D*). $B(t)$ is plotted for each laminar compartment in *Figure 2E*, confirming that border ownership modulation rises earlier in the deep layers than in the other laminar compartments. We quantified the difference by defining latencies on these functions as the crossing with a threshold defined from the null distribution obtained by shuffling the stimulus labels (see Methods). Latency was significantly shorter for deep layer units (75.8 ms, 95% CI [68.4 85.2]) than for granular layer units (94.7 ms, 95% CI [82.2 105.7], bootstrap procedure [see Methods] p = 0.006) and for superficial layer units (97.7 ms, 95% CI [78.0 103.7], bootstrap procedure p = 0.018). The same was true when well-isolated units and multiunits were pooled (deep: 78.2 ms, 95% CI [73.3 88.2], *n* = 210; granular: 94.0 ms, 95% CI [87.9 106.5], *n* = 167; superficial: 100.7 ms, 95% CI [85.1 104.6], *n* = 140; deep vs. granular: p = 0.007; deep vs. superficial: p = 0.009). To verify the robustness of the temporal differences between layers, we also evaluated the time course of border ownership selectivity using another method, by evaluating border ownership reliability (*BOR*) (*Figure 2F*; introduced by *Zhou et al., 2000*; detailed in Methods). Briefly, this metric quantifies the reliability of border ownership tuning when comparing spike counts between single trials to stimuli with opposite border ownership. Reliability values correspond to the proportion of such single trial comparisons for which the spike count is highest for the border ownership condition that is preferred across trials. We computed *BOR* in 100-ms sliding windows (*Figure 2F*; latency defined similarly as in *Figure 2E*, using right edge of the analysis window). Again we find that *BOR* rises earlier in the deep layers (89.9 ms, 95% CI [81.5 99.9]) than in the granular (105.4 ms, 95% CI [94.9 114.6]; bootstrap procedure p = 0.015) and superficial layers (109.4, 95% CI [98.9 124.9]; p = 0.006). After border ownership selectivity has been established, the average border ownership index tends to be higher in the deep compartment (*Figure 2E*; *BOI* between 200 and 500 ms after stimulus onset: mean ± standard error of the mean [SEM], 0.50 ± 0.02) than in the granular (0.43 ± 0.02) and superficial compartment (0.45 ± 0.03), but the differences do not reach statistical significance (deep vs. granular: Wilcoxon rank sum test p = 0.051; deep vs. superficial: p = 0.22). *BOR* saturates around 0.85 in all three compartments (*Figure 2F*). Together, these data indicate that border ownership selectivity does not occur first in granular layer units but instead in deep layer units.

To test whether the short latency in deep layers is specific for border ownership or a general feature of this laminar circuit, we performed two additional analyses. First, we evaluated the latency of spikes evoked by small ring stimuli in the cRF. For these responses we find, in contrast to border ownership selectivity, that the latency is shorter in the granular layer compared to the deep layers and superficial layers (*Figure 3*; latency defined as crossing of the functions with a threshold value a third of the way from baseline to peak). Note that these responses are derived from the same stimuli used to compute the CSD maps, but represent a different signal (spiking responses as opposed to

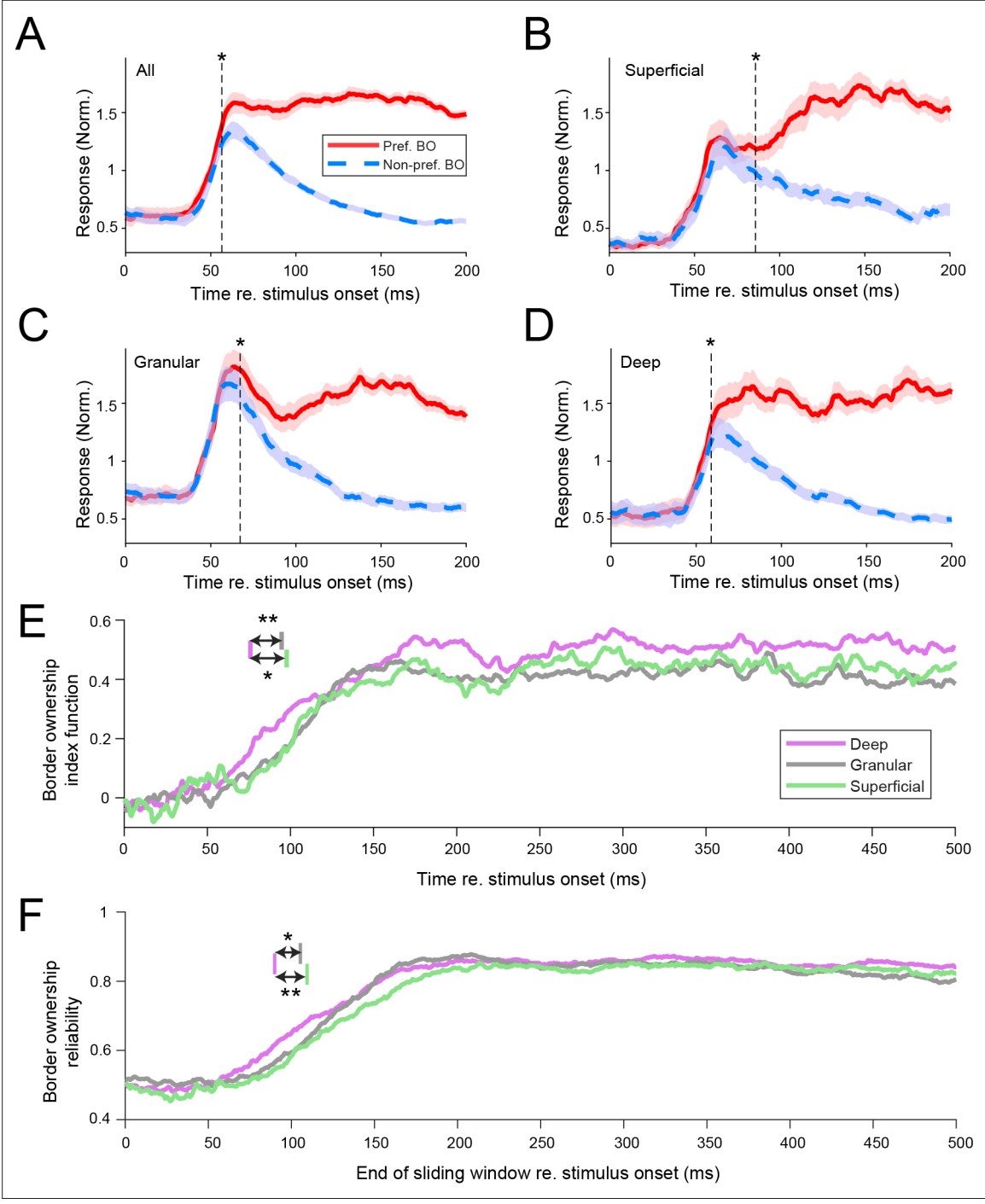

**Figure 2.** Border ownership selectivity occurs first in deep cortical layers. (**A**) Response time courses for the population of border ownership-selective well-isolated units (*n* = 350). Functions are plotted separately for the responses to the preferred side of border ownership (solid red line; mean ± standard error of the mean [SEM]) and the non-preferred side of border ownership (dashed blue line). The vertical dashed line indicates when the difference between the functions first becomes significant (Wilcoxon signed rank test, p < 0.05 for ≥ 20 ms). (**B–D**) Similar to A, for the subpopulations of well-isolated units selective for border ownership that could be located, respectively, to superficial (**B**), granular (**C**), and deep (**D**) layers (see Methods for criteria of layer assignment). Superficial: *n* = 58 units; granular: *n* = 102 units; deep: *n* = 121 units. Analysis as in panel A. See also *Figure 2—figure supplement 1*. (**E**) Border ownership index functions of the different laminar compartments (see Methods). Colored vertical lines indicate latency for the different layers, defined as the earliest crossing of the border ownership index function for ≥20 ms with the threshold. The threshold was set at the level for which <1% of functions obtained after shuffling the stimulus labels had a defined latency (0.156). Latency, deep: 75.8 ms, 95% CI [68.4 85.2]; granular: 94.7 ms, 95% CI [82.2 105.7]; superficial: 97.7 ms, 95% CI [78.0 103.7]. **Bootstrap procedure (see Methods) p = 0.006; *p = 0.018. See also *Figure 2— figure supplements 2 and 3*. (**F**) Border ownership reliability calculated in a 100 ms sliding window for the three laminar compartments. Colors as in E.

*Figure 2 continued on next page*

*Figure 2 continued*

Colored lines in top of panel indicate the earliest crossing for ≥20 adjacent ms with the threshold, defined similarly as for panel E. Threshold crossings: deep 89.9 ms, 95% CI [81.5 99.9]; granular 105.4 ms, 95% CI [94.9 114.6]; superficial 109.4 ms, 95% CI [98.9 124.9]. **Bootstrap procedure p = 0.006; *p = 0.015.

The online version of this article includes the following figure supplement(s) for figure 2:

**Figure supplement 1.** Differences in sample size do not explain differences in latency between layers.

**Figure supplement 2.** Selectivity for contrast polarity does not occur earliest in the deep layers.

**Figure supplement 3.** Laminar analysis is not affected by how the current source density (CSD) maps were smoothed.

the current sink–source patterns from local field potentials used to define the laminar compartments). Second, we evaluated the responses to the border ownership stimuli for another type of selectivity, contrast polarity. This refers to the relative luminance contrast across the edge, that is the difference between panels 1 and 3 (or between panels 2 and 4) in *Figure 1A*. For the contrast polarity index functions (*Figure 2—figure supplement 2*), we do not find an earlier rise in selectivity in the deep layers, even though they are derived from the same units as in *Figure 2*. Together, these data indicate that the earlier selectivity in deep layers compared to granular and superficial layers is specific for border ownership.

## The preferred side of border ownership is organized in columnar clusters

How does the preferred side of border ownership compare between units recorded in a column of cortex? For a given edge, for example vertical, there are two possibilities for preferred side of border ownership: left or right from the edge. Border ownership for such an edge has been assumed to be represented by the activity of two oppositely tuned subpopulations (green and purple in *Figure 4A*, top; *Craft et al., 2007*). Our data show indeed that these two subpopulations exist in similar proportions (*Figure 4A*, top). The same is true for units that encode border ownership for horizontal edges (*Figure 4A*, bottom), and when we express the preferred side of border ownership not relative to the screen, but relative to the fixation point (*Figure 4—figure supplement 1*). This indicates that for a given edge, the two possible sides of border ownership are encoded by distinct subpopulations of neurons that are similar in size.

These two subpopulations could be mixed in a salt-and-pepper pattern, or they could be clustered according to their preferred side of border ownership (*Figure 4B*). *Figure 4C* shows the signed *BOI*

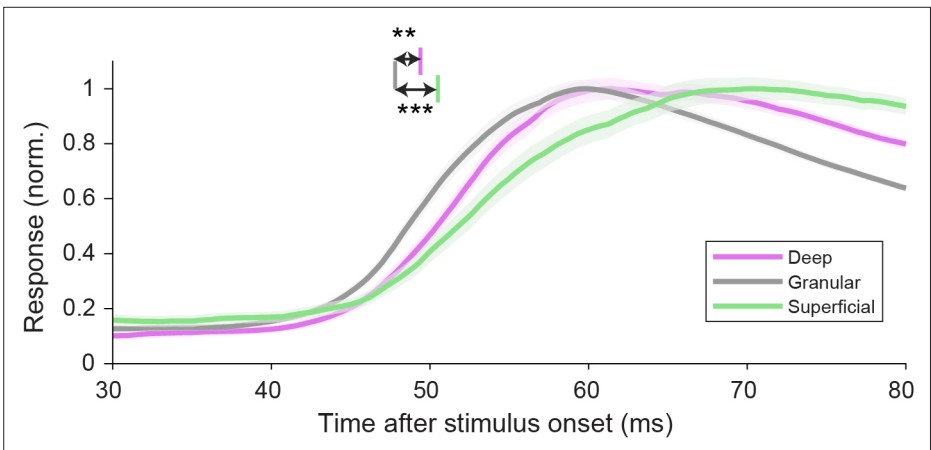

**Figure 3.** Spike latencies evoked by flashed stimuli appearing in the classical receptive field are shortest in the granular layer. Response time courses of spikes evoked by small rings centered on the classical receptive field, for well-isolated units recorded in each laminar compartment (mean ± standard error of the mean [SEM]). Vertical lines indicate mean latencies for each laminar compartment (defined as crossing of the functions with a threshold value a third of the way from baseline to peak, 0.435). Deep: 49.4 ms, 95% CI [48.3 50.3], 173 units; granular: 47.8 ms, 95% CI [47.2 48.5], 152 units; superficial: 50.5 ms, 95% CI [49.1 52.2], 101 units. ***Bootstrap procedure p = 0.0005; **p = 0.002.

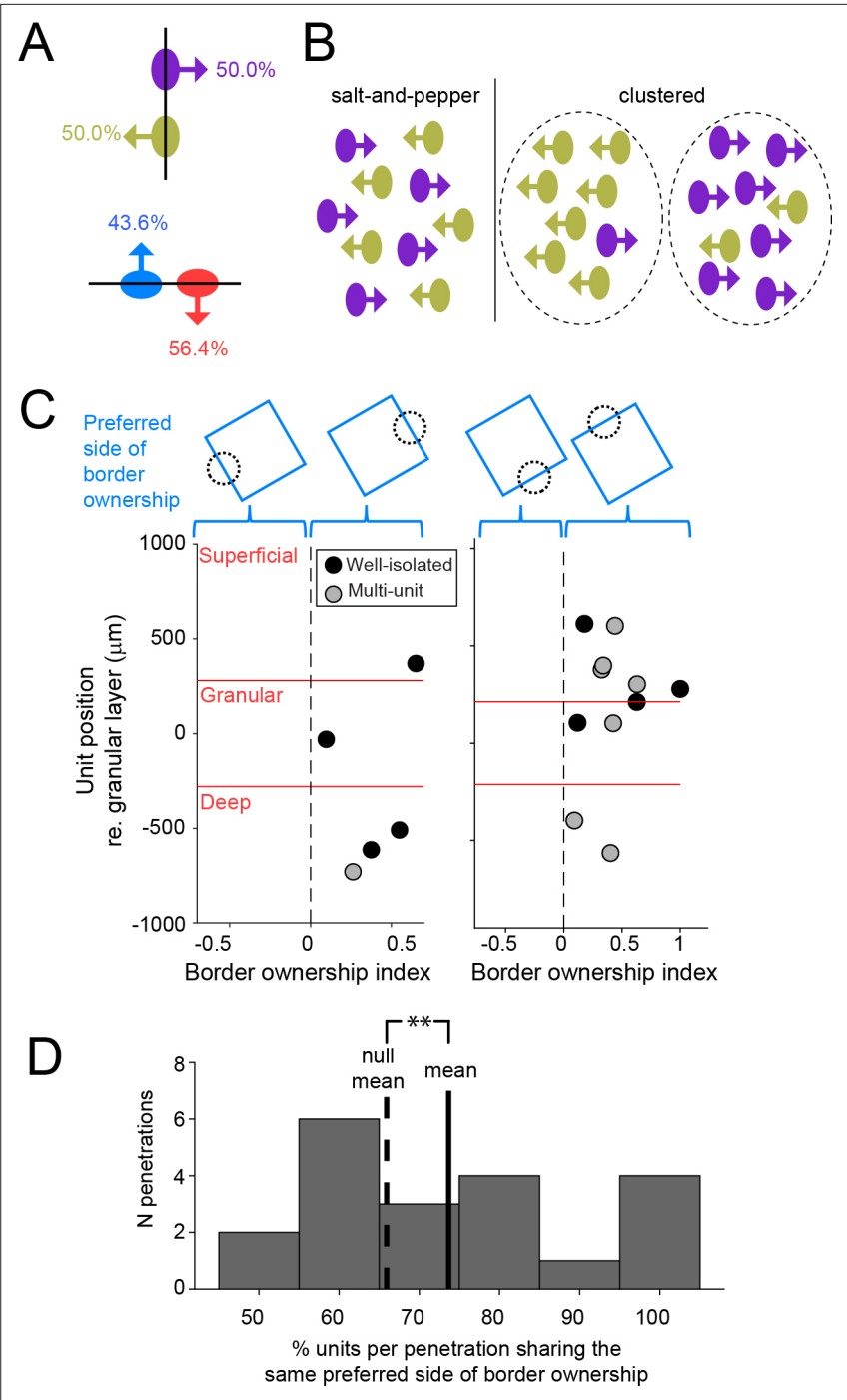

**Figure 4.** Preferred side of border ownership is clustered. (**A**) For a given edge, there are two possibilities for preferred side of border ownership. Percentages indicate the fraction of well-isolated units selective for border ownership that have the preferred side of border ownership indicated by the arrow (vertical edge: *n* = 64; horizontal edge: *n* = 78). See also *Figure 4—figure supplement 1*. (**B**) Cartoon showing possibilities for spatial organization of neurons with opposite preferred sides of border ownership, for a vertical edge. (**C**) Each panel shows border ownership-selective units recorded during one penetration. The abscissa shows the signed *BOI* of each unit (sign indicates preferred side of border ownership, shown above the panels). Ordinate shows the position of the units relative to the center of the granular layer. (**D**) Histogram of the proportion of units per penetration that shares the same preferred side of border ownership, for all penetrations with at least four border ownership-selective well-isolated and multiunits spanning from superficial to deep cortical layers. Solid line is distribution mean, dashed line shows null distribution mean. **\*\*p = 0.003 (20 penetrations with 136 units).

*Figure 4 continued on next page*

*Figure 4 continued*

The online version of this article includes the following figure supplement(s) for figure 4:

**Figure supplement 1.** Similar to *Figure 4A*, but expressing preferred side of border ownership relative to the position of the fixation point.

for all well-isolated units and multiunit clusters selective for border ownership for two example penetrations. The sign of *BOI* indicates which side of the border is preferred by each unit, as indicated by the cartoons above the panels. For both of these penetrations, all border ownership units recorded on the same probe prefer the same side of border ownership. In the population of all penetrations with at least four border ownership units, the proportion of units that share the same preferred side of border ownership is significantly higher than chance (well-isolated units: 74.4%, randomization test p = 0.017, 23 penetrations with 110 units; including multiunit clusters: 71.7%, p = 0.005, 47 penetrations with 288 units). Also in the subgroup of penetrations with border ownership units spanning from superficial to deep layers, we find significant clustering (*Figure 4D*; well-isolated units: 81.1%, randomization test p = 0.010, 6 penetrations with 30 units; including multiunit clusters: 73.7%, p = 0.003, 20 penetrations with 136 units). Together, these data indicate that border ownership preference is organized in spatial clusters that span cortical layers in a columnar fashion.

## Columnar organization of orientation selectivity

Border ownership acts on edges that are oriented. Similar to our finding that preferred border ownership is clustered, several imaging studies have shown that preferred orientation is spatially clustered in V4 in domains (e.g., *Li et al., 2013*; *Tanigawa et al., 2010*), but it is not known whether these form columns. To test whether orientation preference in V4 is shared in clusters that extend vertically across layers, we determined orientation tuning for the units in our sample from responses to luminance contrast edges centered on the cRF (independent data set from the border ownership stimuli, see Methods). Our data confirm clustering of preferred orientation and reveal that these clusters span across laminar compartments in V4 (*Figure 5*). Each polar plot in *Figure 5* shows a penetration with at least four orientation-selective well-isolated units or multiunit clusters. Solid vectors indicate the preferred orientation of each orientation-selective unit as the resultant vector of the responses to edges of different orientations, and color indicates laminar compartment (open symbols indicate multiunit clusters). The preferred orientation for a penetration was then calculated as the resultant vector across the vectors of all orientation-selective units in that penetration, and its angle is indicated by the blue dashed line. The significance of clustering of orientation preference was assessed by comparing the magnitude of this resultant vector against a null distribution generated by randomizing the preferred orientation for each orientation-selective unit in the penetration and performing the same calculation. For each of these six penetrations, this resultant was significantly larger than expected from the null distribution (p < 0.05). This was true for 9 out of 18 penetrations with at least four orientation-selective well-isolated units distributed over all three laminar compartments (for 17 out of 34 penetrations when well-isolated units and multiunit clusters were pooled). These data suggest that orientation domains in V4 are columnar.

## Border ownership selectivity often occurs far away from the preferred orientation

Having found that both border ownership preference and orientation preference are organized as columnar clusters in V4, we next asked what the relation was between orientation selectivity and border ownership selectivity. We find that border ownership selectivity is significantly more common in units that are selective for orientation than in those that are not, but it is also surprisingly common in units that are not selective for orientation (*Table 1*, respectively, 53.3% and 36.8%, Chi square = 9.51, p = 0.002; including multiunit clusters: respectively, 45.9% and 31.3%, Chi square = 19.5, p = 0.00001).

Three example orientation-selective well-isolated units are shown in *Figure 6A*. These polar plots are organized according to the location of the square object relative to the cRF, for different orientations of the central edge (the edge that runs through the cRF). This is indicated by the stimulus cartoons around the plot. For opposite locations on the polar plot, the central edge has thus the same

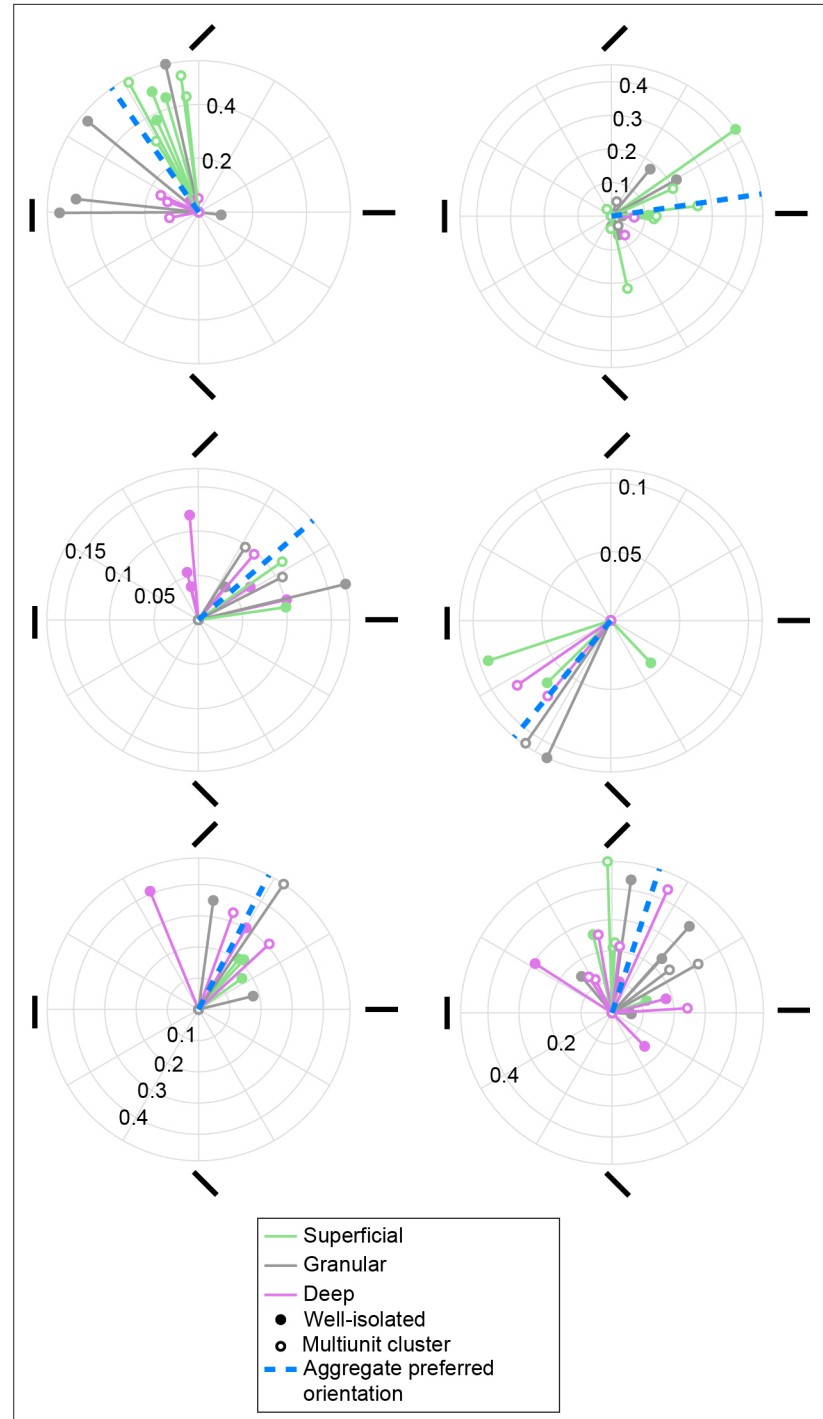

**Figure 5.** Columnar clusters of preferred orientation in V4. Polar plots are shown for six penetrations with at least four orientation-selective units distributed across all three laminar compartments. Each solid vector corresponds to one unit and indicates the preferred orientation (direction) and the degree of orientation tuning (magnitude), quantified, respectively, as the direction and magnitude of the resultant vector of the responses to each orientation. Color indicates the laminar compartment. Open symbols indicate multiunits, closed symbols indicate well-isolated units. Blue dashed line indicates the preferred orientation across units in a penetration (aggregate preferred orientation). All six penetrations shown have a significant aggregate preferred orientation.

**Table 1.** Border ownership selectivity is more common in orientation-selective well-isolated units but often occurs in units that are not selective for orientation.

| | | Border ownership selectivity | |
|---|---|---|---|
| | | − | + |
| Orientation selectivity | − | 79 | 46 |
| | + | 133 | 152 |

orientation, but is owned by a square positioned on opposite sides of the cRF (opposite border ownership). The *BOI* for each tested orientation of the central edge is shown by a red vector (filled red circles indicate statistically significant border ownership selectivity). The amplitude of this vector corresponds to |*BOI*|, and the polarity of the vector points toward the square location that corresponds to the preferred side of border ownership for that orientation of the central edge. For example, in case of a vertical central edge, the third unit (cyan triangle) prefers that that edge is owned by a square on the right (if it would have preferred the vertical central edge to be owned by a square on the left, the red vector that points down would have pointed up). The black line indicates the preferred orientation of a luminance contrast edge in the cRF for each unit. For the left unit in *Figure 6A* (red star), this preferred edge orientation matches with the orientation of the central edge for which border ownership selectivity is significant (filled red circle). This is the expected scenario: border ownership has been assumed to act on edges at the preferred orientation, and prior studies have used edges at the preferred orientation to test border ownership selectivity (*Zhou et al., 2000*). At the population level, the average orientation of edges with border ownership selectivity is indeed mildly biased toward the preferred orientation for edges. This is subtle but can be seen as the tendency of the data points in *Figure 6C* to appear close to the identity lines. Note that both plotted variables in *Figure 6C* are periodic and two periods are shown for each (thus each data point is plotted four times, and the identity line appears three times; the gray square outlines an area corresponding to one period for both variables). This bias is easier to see when the closest distance of the data points to the identity lines is plotted (*Figure 6D*). This distance is on average distance 27.6°, significantly smaller than chance (randomization test [see Materials and methods] p = 0.018; n = 68 well-isolated units tested with at least four orientations; including multiunit clusters: p = 0.017; n = 159). Orientation-tuned units thus tend to be border ownership-selective for edges with an orientation near their preferred orientation, but this relation is surprisingly subtle.

Some units only have border ownership selectivity for orientations that are far from the identity line, such as the unit indicated with the blue square in *Figure 6A* (middle panel, corresponding to the same symbol in *Figure 6C*). This unit shows, paradoxically, significant border ownership selectivity for edge orientations that are nearly orthogonal to the preferred edge orientation. This is true for two edge orientations (filled red circles), suggesting that this misalignment is systematic. We ruled out that this is related to a difference in orientation preference for isolated edges versus edges that are part of squares by comparing orientation tuning for both data sets. *Figure 6—figure supplement 1A* shows that the preferred orientations for both stimuli match very well for this unit (black line vs. cyan line), and this is true for the population as well (*Figure 6—figure supplement 1B*).

Furthermore, we find that border ownership-selective units often show border ownership selectivity to edges that maximally differ in orientation, that is orthogonal edges. An example unit is shown in the right panel in *Figure 6A* (cyan triangle). This is true for 29.7 % of border ownership-selective well-isolated units (n = 182; 24.4 % including multiunits, n = 353) that were tested with sets of squares at orthogonal angles. In such cases, the preferred side of border ownership for a central edge with an orientation in between those orthogonal orientations could in theory be spatially discontiguous (*Figure 6B*, left panel) or contiguous (*Figure 6B*, middle panel) with the preferred sides of border ownership for the pair of orthogonal orientations. We find that for all 24 out of 24 well-isolated units that were selective for the border ownership of squares at orthogonal and intermediate orientations, the preferred side of border ownership for these intermediate orientations was spatially contiguous (such as the example in *Figure 6B*, right panel; including multiunits, this is true for 35 out of 35 units). Those units thus have a wide but spatially contiguous area of preferred sides of border ownership. For example, the unit in *Figure 6B* (right panel) has an area of preferred sides of border ownership that spans 120° (cyan double arrow). A span of at least this width occurs for 18.0 % of border ownership units (*Figure 6E*; 15.4 % including multiunit clusters).

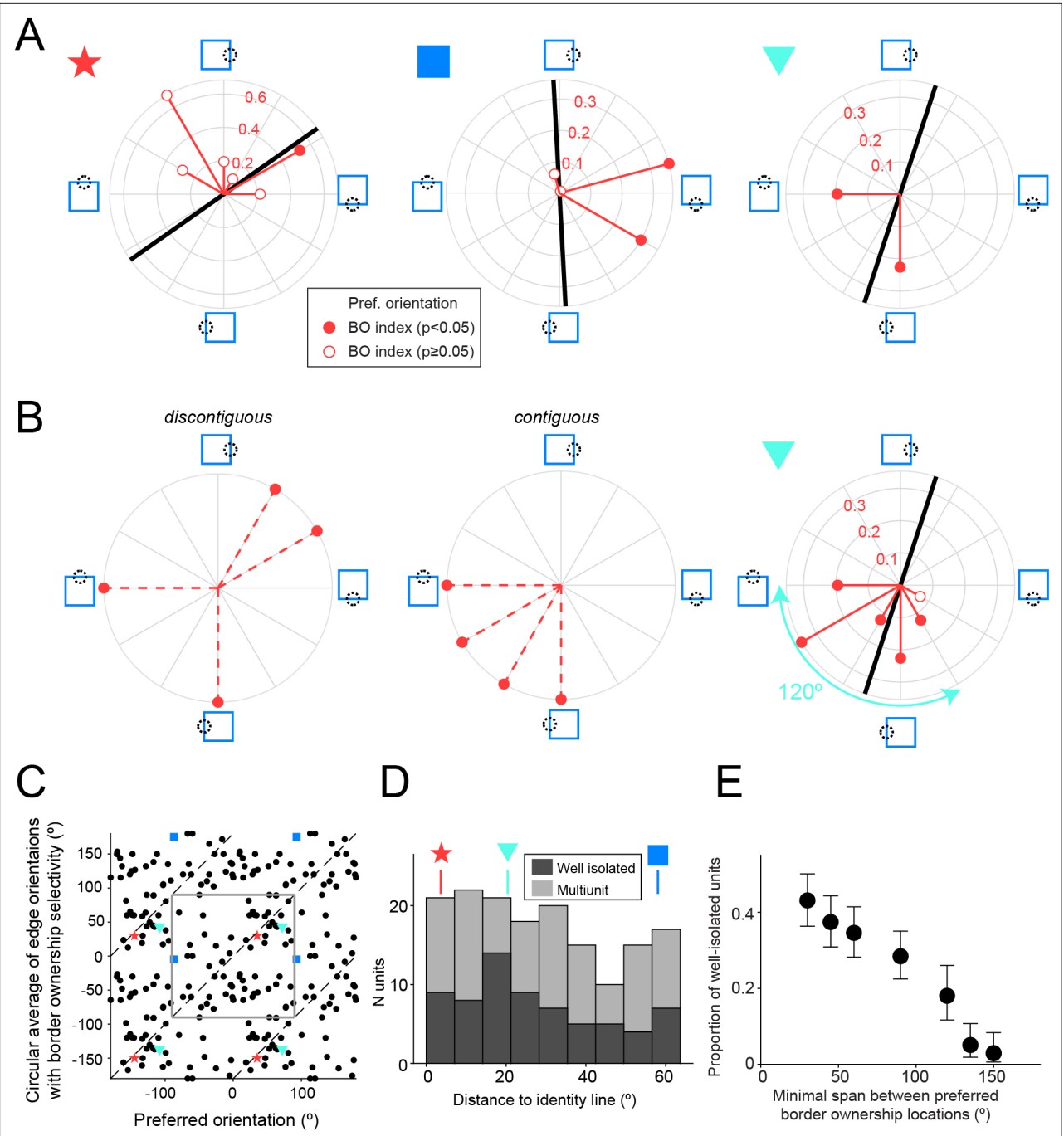

**Figure 6.** Border ownership selectivity often occurs far away from the preferred orientation. (**A**) Polar plots for three example units that were tested for border ownership selectivity with edges of multiple orientations (orientations of red vectors indicate tested orientations). Polarity of red vector relative to origin indicates the preferred side of border ownership for each edge orientation, according to the stimulus cartoons surrounding the plot, and filled circles indicate cases with statistically significant border ownership selectivity (adjusted for multiple comparisons using Bonferroni correction). Magnitude of red vectors corresponds to |*BOI*|. Black line indicates preferred edge orientation. Colored symbols on the upper left side of each polar plot correspond to the symbols in C and D. See also *Figure 6—figure supplement 1*. (**B**) Left and middle: for units showing border ownership selectivity to orthogonal edges there are two possibilities for the intermediate edge orientations. Either the preferred sides of border ownership are spatially discontiguous with those for the orthogonal edges (left), or they are contiguous (middle). Right: polar plot for the same unit as in A (right polar plot), now including data for the orientations that were not shown in A. (**C**) Scatter plot comparing the preferred orientation for edges (abscissa) with the circular mean of edge orientations for which border ownership selectivity is significant (ordinate), for all units that were tested for border ownership for at least four orientations (*n* = 68 well-isolated units). Note that each data point is plotted four times, because both variables are periodic and two periods are shown for each. The gray square outlines an area corresponding to one period for both variables. Dashed lines indicate identical

*Figure 6 continued on next page*

*Figure 6 continued*

values. Colored symbols correspond to the units in A, B, and D. (**D**) Smallest orthogonal distance to the identity line for the units shown in C. The data are significantly biased toward zero (see Results). Distance values for the example units shown by colored symbols in A–C are indicated above the histogram. (**E**) Proportion of border ownership well-isolated units where the preferred sides of border ownership cover an angular span at least as wide as the value indicated along the abscissa (for all units tested with such spans). For example, the unit in B (right) has a span of preferred sides of border ownership of 120°. Note that the span between spatially contiguous preferred sides of border ownership is necessarily <180°. *N* units tested for each value of on the abscissa: ≥30°: 211; ≥45°: 211; ≥60°: 211; ≥90°: 211; ≥120°: 122; ≥135°: 118; ≥150°: 102.

The online version of this article includes the following figure supplement(s) for figure 6:

**Figure supplement 1.** Consistent orientation tuning for isolated edges and for edges that are part of squares.

## Discussion

The correct assignment of borders is a key step in the process of constructive perception that enables us to identify and localize objects in a visual scene. Prior studies have found neurons that signal border ownership, but the underlying circuitry is not understood. Various theoretical accounts have been offered. One group of models hypothesizes that border ownership selectivity arises in a feedforward manner, through the successive elaboration of progressively more complex receptive field properties (*Sakai et al., 2012*; *Sakai and Nishimura, 2006*; *Supèr et al., 2010*). Other accounts posit that border ownership assignment crucially depends on horizontal connections and corticocortical feedback signaling (*Craft et al., 2007*; *Grossberg, 2015*; *von der Heydt, 2015*; *Zhang and von der Heydt, 2010*; *Zhaoping, 2005*). These different pathways differ in their laminar organization. Feedforward inputs terminate mostly in the granular layer, whereas long-range horizontal fibers and cortical feedback projections avoid the granular layer (*Douglas and Martin, 2004*; *Rockland and Pandya, 1979*; *Rockland et al., 1994*; *Rockland and Lund, 1983*; *Ungerleider and Desimone, 1986*; *Yoshioka et al., 1992*).

### Border ownership selectivity and the laminar circuit

Given that about half of the neurons responsive to edges in V2 are selective for border ownership (*Zhou et al., 2000*), neurons in V4 could simply inherit this property from V2 through the feedforward projection from V2 to V4. This projection is the most important source of feedforward input to V4 (*Markov et al., 2011*), and targets mostly layer 4 (granular layer) in V4 (*Gattass et al., 1997*; *Rockland, 1992*). Our data show that spike latency for stimuli in the cRF is shortest in the granular layer, which is similar to what has been described in other areas in macaque visual cortex (V1: e.g., *Nowak et al., 1995*; MT: *Raiguel et al., 1999*), and consistent with work by others in awake V4 (*Lu et al., 2018*). Note that this may not be universal across areas: in anesthetized V2 spike latency has been reported to be longer in layer 4 than in infragranular layers (*Nowak et al., 1995*).

If this feedforward projection would thus simply provide the earliest border ownership signals in V4, we would thus detect border ownership selectivity in the granular layer at least as early as in the other compartments, or earlier. While we do observe such a pattern for responses evoked by flashed stimuli appearing within the cRF, and also for the emergence of selectivity for contrast polarity (*Figure 2—figure supplement 2*), we do not observe this pattern for selectivity to border ownership. Instead, we find that deep layer neurons compute border ownership selectivity significantly earlier than neurons in the granular layer and in the superficial layers. One might argue that these data are consistent with V2 afferents preferentially targeting apical dendrites of deep layer neurons that extend in the input layer. But in that case, we would expect shorter spike latencies for deep layer neurons than for granular layer neurons irrespective of the cue or stimulus. This is, again, not what we find: spike latencies for responses evoked by small stimuli in the cRF are shorter in the granular layer than in the deep layers, and also for contrast polarity we do not observe the earliest selectivity in deep layers in V4.

Another possibility is that deep layer neurons perform this computation earliest solely by relying on an integration of feedforward signals from the V2-to-V4 projection relayed by multiple granular layer neurons. We think this is unlikely. First, this would mean that those relaying granular layer neurons do not receive sufficient contextual information to become selective for border ownership as fast as the deep layer neurons that integrate their output, despite the substantial input that granular layer neurons receive from other granular layer neurons (*Xu et al., 2016*). Second, while deep layer neurons certainly receive input from granular layer neurons, the densest projections from granular

layer neurons typically target superficial layers (e.g., macaque V1: ; *Callaway, 1998*; rat barrel cortex: *Lübke et al., 2000*). If that projection leads to the first computation of border ownership signals, one would thus expect to observe these signals in superficial layer neurons as fast as in deep layer neurons, in contrast with our data.

We think that the earlier computation of border ownership selectivity by neurons in deep layers is more likely a consequence of their unique properties. Scanning laser photostimulation studies in primary visual cortex showed that layer 5 neurons receive significant input from nearly all cortical layers, regardless of cell type, as opposed to neurons in other layers (*Briggs and Callaway, 2005*; *Xu et al., 2016*). Deep layers of the cortex, including in primates, include tall pyramidal cells whose apical dendrites reach up to layer 1, which could thus directly sample and integrate afferent information that arrives in a wide range of layers (*Binzegger et al., 2004*; *Callaway, 1998*; *Lund and Boothe, 1975*; *Markov et al., 2014*; *Zarrinpar and Callaway, 2016*). Such neurons thus seem well suited to integrate information provided by feedforward input with contextual information provided through corticocortical feedback and horizontal connections (*Harris and Shepherd, 2015*). Corticocortical feedback in V4 terminates densely in all layers except layer 4 (*Markov et al., 2011*; *Rockland et al., 1994*). Intra-areal horizontal connections are prominent in layer 2/3 and in layer 5 (*Lund et al., 1993*; *Yoshioka et al., 1992*; *Douglas and Martin, 2007*). This may set these deep layer neurons up to be able to integrate the border in the cRF (light gray arrow in *Figure 7*) with visual context from a wide region of space (dark gray arrows in *Figure 7*), which is required to compute border ownership selectivity. Indeed, studies in V2 showed that border ownership selectivity does not rely on a small number of localized object features but instead occurs through integration of extraclassical stimulus features over large areas of visual space (*Zhang and von der Heydt, 2010*). Specialized intrinsic properties could further assist in this integration, such as the calcium spikes in apical dendrites of layer five neurons that can amplify the effects of feedback inputs (*Takahashi et al., 2016*). It would be interesting to test whether deep layer neurons in other areas, for example in V2, also display early border ownership signals (although obtaining penetrations orthogonal to the cortical surface in macaque V2 is challenging, given that only a small part of this area is exposed on the lateral convexity of the brain). Perhaps such contextual integration is a general computation performed by deep layer neurons in different cortical areas. In addition, as discussed below, deep layer neurons in V2 may inherit early border ownership signals from V4.

Our findings are consistent with the concept of the grouping cell model (*von der Heydt, 2015*). This model proposes that border ownership is a modulation of feature neurons by an external grouping signal that acts through feedback. The grouping signal is supposed to represent a 'proto-object', a computational structure that (1) links the visual feature signals, (2) is being remapped across eye movements, and (3) serves object-selective attention. Compared to the large number of studies of feature selectivity in visual cortex, the question of the representation of object structure has prompted relatively few studies. It remains unclear where and how such grouping signals are generated (*Zhu et al., 2020*). It is also possible that these deep layer neurons in V4, rather than computing border ownership selectivity de novo from an external grouping signal, inherit this selectivity from higher areas through corticocortical feedback. The source of feedback, whether it provides border ownership signals or grouping signals, would need to have a spike latency that is short enough to explain the early border ownership signals in deep layers in V4. Because of shorter latencies, dorsal stream areas such as LIP may therefore be more likely sources of such feedback than inferotemporal cortex, where shape-selective signals occur after 100 ms (*Brincat and Connor, 2006*; *Bullier, 2001*; see also discussion in *Zhu et al., 2020*).

Our conclusion that the earliest border ownership signals in V4 are not inherited from V2 does not imply that the projection from V2 to V4 does not carry any border ownership signals. Indeed, since about half of V2 neurons are selective for border ownership (*Zhou et al., 2000*), this feedforward input most likely contributes to the border ownership signals in V4 later in the response. As argued in the next section, these border ownership signals in V2 may have been sculpted by border ownership-selective feedback from deep layers in V4. That V4 does not simply inherit its earliest border ownership signals from V2 is also consistent with a study in V4 on shape encoding that compared responses to identical shapes in the presence or absence of an occluder (*Bushnell et al., 2011*). When the preferred shapes had sharp convexities and concavities, responses were suppressed if those features were rendered incidentally by an adjacent occluder. Suppression latency was relatively short (63 ms after stimulus

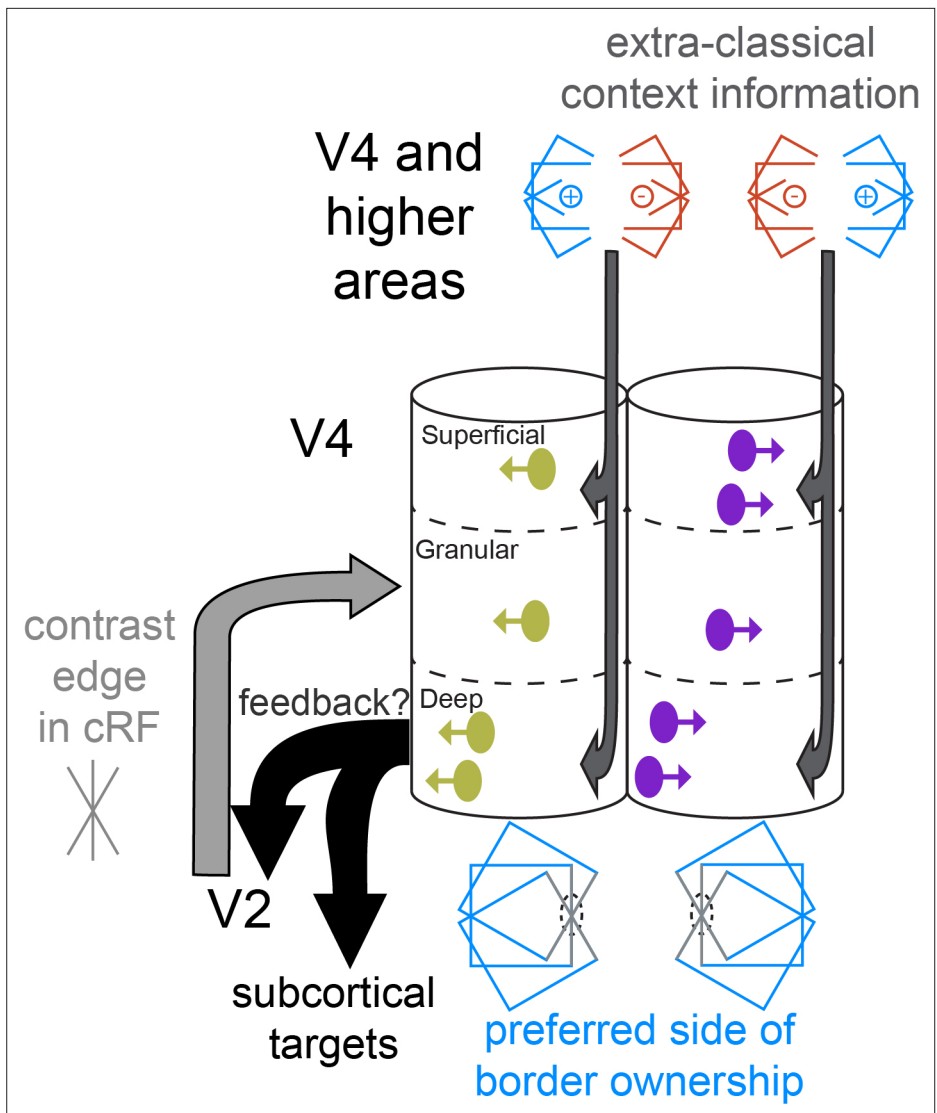

**Figure 7.** Cartoon of a conceptual model of columnar processing of border ownership, supported by our data. Two columnar structures are shown, each of which responds to vertical edges in the classical receptive field (cRF), but they have opposite preferred sides of border ownership (indicated by the direction of the arrows on the units in the columns, and by the symbol below each column). Such columns may or may not be adjacent. The earliest occurrence of border ownership selectivity in the deep layers is symbolized by the positon of deep layer units near the left edge of the column. Light and dark gray arrows indicate where different types of stimulus information likely arrive first. Stimulus information in the cRF (gray edge) arrives first in the granular layer. Deep layer neurons may first compute border ownership selectivity by integrating this information with contextual information that arrives in superficial and deep layers, possibly provided by corticocortical feedback and horizontal connections. The content of these context signals is necessarily asymmetric to result in border ownership preference, and different for the two columns shown (indicated by the symbols above the columns). Because of their prominence in deep layers, early border ownership signals may be part of the feedback that projects from V4 to upstream areas (e.g., V2), and to subcortical targets, such as the superior colliculus (black arrows). Later on during the response, after border ownership signals have first been established in V4, the feedforward input from V2 to V4 likely also contains border ownership signals, since roughly half of V2 units are selective for border ownership (*Zhou et al., 2000*).

onset in the population, and for an example neuron as short as 46 ms). Because this is shorter than the latency of border ownership signals reported in V2 (under somewhat different stimulus consitions: ~68 ms; *Zhou et al., 2000*), the suppression effect is likely not due to border ownership signals that are inherited from V2. A caveat when interpreting this suppression as border ownership selectivity is that both the shape and the occluder fell in the cRF. Thus, while the suppression may be due to the

change in border ownership, it may instead be due to local contour geometry at the T-junctions in the cRF. In any case, this suppression of incidental shapes in V4 could contribute to border ownership signals in V2 in the presence of occluders, through feedback.

A computational model has shown that contextual information carried by myelinated feedback afferents could indeed arrive fast enough to result in border ownership selectivity within ~20 ms after response onset (*Craft et al., 2007*). An alternative mechanism based solely on horizontal fibers is unlikely, given the slow conduction velocity of these fibers (*Girard et al., 2001*; *Zhang and von der Heydt, 2010*), although, as discussed by *Zhaoping, 2005*, it should be noted that data on horizontal fibers are scarce. That said, horizontal fibers could still provide part of the required contextual information: since these fibers are limited in length, their role could be to provide information from the portions of the stimulus closest to the cRF (reminiscent to their proposed role in surround suppression in V1 [*Angelucci et al., 2017*]). This would explain why the effect of near corners of squares tends to occur later than that of far corners of squares (*Zhang and von der Heydt, 2010*).

## Potential function of early border ownership signals in deep layers

Once border ownership selectivity has been established in V4 deep layers, these signals may contribute to visual processing along different pathways (black arrows in *Figure 7*). First, V4 is the dominant source of cortical feedback to V2, and 75 % of these feedback neurons are located in deep layers in V4 (*Markov et al., 2014*). Early border ownership signals in deep layers of V4 may thus sculpt border ownership selectivity in V2. Second, V4 deep layers include neurons that project to the superior colliculus (*Fries, 1984*; *Gattass et al., 2014*). The superior colliculus may thus receive early border ownership signals from V4, which could contribute to planning upcoming saccades. Saccades target objects more frequently than background (*Rothkopf et al., 2007*). Early border ownership signals in deep layer neurons that project to the superior colliculus could serve to facilitate the rapid foveation of objects. Our data show that these signals are established in deep layers well before 100 ms after stimulus onset. They are thus computed quickly enough to allow time for saccade planning within the intersaccadic interval, which is typically on the order of 200–300 ms (*Otero-Millan et al., 2008*). This proposed early role for deep layer neurons in supporting saccadic decisions is consistent with recent findings showing that deep layer V4 neurons encode more information about the direction of planned eye movements than do superficial neurons (*Pettine et al., 2019*). Border ownership signals may thus play a role beyond visual perception, perhaps including a role in guiding rapid oculomotor behavior. Finally, deep layer neurons may convey border ownership signals to neurons in more superficial layers, which would be consistent with the canonical laminar circuit (*Douglas and Martin, 2004*). Our results suggest a potential strategy to study the flow of these signals within single trials. We find that border ownership preference is shared across layers in a columnar fashion. Border ownership is presumably evaluated in the brain by comparing the activity between neurons from such oppositely tuned border ownership columns. In future work, it would thus be interesting to record from such columns simultaneously, and use directionality measures such as Granger causality (e.g., *Ferro et al., 2021*) to study the interlaminar flow of border ownership signals within single trials.

## Border ownership and orientation tuning

Prior studies on border ownership only evaluated border ownership selectivity at the preferred orientation of each neuron (*Hesse and Tsao, 2016*; *Zhang and von der Heydt, 2010*; *Zhou et al., 2000*). In our data, there is indeed a bias toward border ownership selectivity for edges near the preferred orientation. However, we find that there is substantial scatter (*Figure 6C and D*). This does not seem to stem from measurement noise. First, there is a good match between preferred orientations estimated from two separate data sets: one based on recordings made with isolated edges, the other using edges that are part of squares (*Figure 6—figure supplement 1*), suggesting that our estimates of orientation preference are highly reliable, with relatively tight bounds. Second, for some units, border ownership selectivity, surprisingly, systematically only occurs for edges with orientations that are nearly orthogonal to the preferred orientation (*Figure 6A*, middle). Third, almost a third of units that are selective for border ownership to an edge with a given orientation are also border ownership-selective to an edge that is orthogonal to that orientation. This reflects systematic tuning, because the preferred sides of border ownership in such cases form a single contiguous area in retinotopic space (*Figure 6B*). Finally, a substantial number of units that are selective for border ownership are not

selective for orientation (*Table 1*). Together, these data indicate that the relation between orientation and border ownership is much richer than has previously been appreciated. Encoding of orientation and of border ownership may represent separate axes of representation. This is not inconsistent with the idea that border ownership assignment represents a surface signal rather than a property tied to a border (*Grossberg, 2015*; *Nakayama et al., 1995*; *Peterson and Skow, 2008*).

## Clustering of border ownership selectivity

Functional clustering is a recurring theme in primate cortical visual areas (*Hubel and Livingstone, 1987*), and has been reported for several modalities in V4 including orientation, hue, direction of motion, spatial frequency, and recently curvature (*Hu et al., 2020*; *Jiang et al., 2021*; *Li et al., 2013*; *Liu et al., 2020*; *Lu et al., 2018*; *Roe et al., 2012*; *Tang et al., 2020*; *Tanigawa et al., 2010*). We find here that border ownership preference is preserved across layers within a vertical penetration, indicating that this modality is also organized in a columnar fashion. Importantly though, border ownership is of a fundamentally different nature than these other modalities: the border ownership of an edge is computed based on stimulus features falling outside the cRF. Combined with our finding that early border ownership signals are computed by deep layer neurons rather than being passed on from upstream areas, this columnar organization has important implications for the functional anatomy. It indicates that there is a systematic asymmetric arrangement of the extraclassical contextual information, which, as argued above, may be provided through horizontal fibers and corticocortical feedback. For example, consider a cluster in which neurons prefer that a vertical edge belongs to an object on the left (left column in *Figure 7*, symbol below left column indicates preferred border ownership). The neurons in this column thus need to receive asymmetric contextual synaptic information favoring the presence of an object to the left of the edge (symbol above the left column in *Figure 7*). Neurons in another cluster (right column in *Figure 7*) will instead prefer that the same edge is part of an object on the right, and these neurons will thus necessarily receive asymmetric contextual information of the opposite polarity (symbol above right column). The clustered architecture of preferred border ownership therefore requires clusters of asymmetric contextual information in cortex, such that opposite polarities of contextual information occur in distinct and complementary clusters. The present data indicate that border ownership selectivity initially does not arrive through the feedforward pathway. They thus suggest that the substrate of these clusters consists of clustered asymmetries in the retinotopic information carried by afferents from horizontal fibers and cortical feedback.

# Materials and methods

**Key resources table**

| Reagent type (species) or resource | Designation | Source or reference | Identifiers | Additional information |
|---|---|---|---|---|
| Strain, strain background (*Macaca mulatta*, male) | Macaque | UC Davis | | |
| Software, algorithm | MATLAB | The Mathworks | | |
| Software, algorithm | MonkeyLogic | *Hwang et al., 2019* | | https://monkeylogic.nimh.nih.gov/; https://www.brown.edu/Research/monkeylogic/ |
| Software, algorithm | CircStat | *Berens, 2009* | | |
| Software, algorithm | SpyKING CIRCUS | *Yger et al., 2018* | | |
| Other | 32-channel multielectrode probe | ATLAS Neuroengineering | E32+ R-100-S1-L10 NT | |

## Animals

We obtained recordings in two male rhesus macaques (*Macaca mulatta*), age 13 years (animal Z) and age 15 years (animal D). Both animals had no prior experimental history and were housed in separate cages in a primate room with up to six animals of the same species. This study was performed in accordance with the recommendations in the Guide for the Care and Use of Laboratory Animals of

the National Institutes of Health. All procedures were approved by the Institutional Animal Care and Use Committee of the Salk Institute for Biological Studies (protocol 14-00014).

## Surgery

Surgical procedures have been described before (*Nandy et al., 2017*). In brief, in a first surgical session a titanium recording chamber was installed in a craniotomy over the prelunate gyrus, according to stereotactic coordinates derived from anatomical MRI scans from each animal (left hemisphere in animal Z, right hemisphere in animal D). In a second surgical session, the dura mater within the chamber was removed, and replaced with a silicone-based optically clear artificial dura, establishing an optical window over dorsal area V4.

## Electrophysiology

At the beginning of a recording session, a sterile insert consisting of a metal ring covered with a plastic membrane on the bottom was lowered in the chamber. The membrane was perforated to allow insertion of probes. The function of the insert is to stabilize the recording site from cardiopulmonary pulsations. A linear multielectrode probe (32-channel single-shaft acute probes, 100 μm electrode pitch [ATLAS Neuroengineering, Leuven, Belgium]) was mounted on the chamber using a hydraulic microdrive on an adjustable x–y stage (MO-972A, Narashige, Japan). The probe was then lowered through the artificial dura over the prelunate gyrus, positioned orthogonally relative to the cortical surface under visual guidance (Zeiss microscope). While monitoring the voltage signals from the electrodes for multiunit activity, the probe was lowered to penetrate the cortical surface. The probe was advanced until multiunit activity was visible on the deepest ~2600 μm of the probe. Then, the probe was retracted typically by several 100 μm to ease dimpling of the cortex. Between recording sessions, probe position was varied (RF eccentricity median 4.87 degrees of visual angle (dva), interquartile range 2.51 dva). Neural signals were recorded extracellularly, filtered and saved using Intan hardware (RHD2132 amplifier chip and RHD2000 amplifier evaluation system, Intan Technologies LLC, Los Angeles, USA) controlled by a Windows computer.

## Stimulus presentation

Visual stimuli were presented using a LED projector, back-projected on a rear-projection screen that was positioned at a distance of 52 cm from the animal's eyes (PROPixx, VPixx Technologies, Saint-Bruno, Canada). The MonkeyLogic software package developed in MATLAB (https://www.brown.edu/Research/monkeylogic/; https://monkeylogic.nimh.nih.gov/; *Hwang et al., 2019*) was used for stimulus presentation, behavioral control and recording of eye position. A photodiode was used to measure stimulus timing. Eye position was continuously monitored with an infrared eye tracking system (ISCAN model ETL-200, Woburn, MA) and eye traces were saved using MonkeyLogic. Trials were aborted if eye position deviated from the fixation point (threshold typically 1 dva radius).

## Receptive field mapping stimuli

At the beginning of each recording session, RF mapping data were obtained using a subspace reverse correlation approach (*Nandy et al., 2017*). Stimuli consisted of static modified Gabors, constructed using square-wave instead of sinusoidal gratings, and dark gray rings (80 % luminance contrast, diameter 2 dva, thickness 0.25 dva). Grating spatial frequency and phase were such that a single contrast edge was visible centrally in the Gaussian window (grating parameters: 6 orientations, 2 contrast polarities, typically 80 % luminance contrast, one half was one of seven colors or grayscale, the other half was always grayscale; window: FWHM 2 dva). The stimuli were presented every 50 or 60 ms while the animal maintained fixation. Each stimulus appeared at a random location selected from a grid sized 25 × 25 dva with 1 dva spacing centered at coordinates [7.5 dva; 7.5 dva] in the appropriate visual quadrant. During the recording session, high-gamma filtered voltage waveforms in response to these stimuli were analyzed to estimate the retinotopy of the probe position, and choose location and size for the border ownership stimuli. Detailed RFs were calculated for each unit offline after spike sorting and used to verify the proper position of the border ownership stimuli.

## Layer assignment

A CSD mapping procedure on evoked local field potentials (LFP) was used to estimate the laminar position of recorded channels (*Nandy et al., 2017*). Briefly, animals maintained fixation while dark gray ring stimuli were flashed (32 ms stimulus duration, 94 % luminance contrast, sized and positioned to fall within the cRF of the probe position). The CSD was calculated as the second spatial derivative of the stimulus-triggered LFP (filtered between 3.3 and 88 Hz) and visualized as spatial maps after smoothing using bicubic (2D) interpolation (*Figure 1—figure supplement 1*; MATLAB function *interp2* with option *cubic*, the spatial dimension was interpolated at a resolution of 10 μm), but the laminar analysis did not critically depend on this particular method of smoothing (*Figure 2—figure supplement 3*). Red regions depict current sinks, blue regions depict current sources. As described in more detail in Results, we observed a consistent pattern between different penetrations, strikingly similar to the current sink–source maps reported by other laboratories in behaving macaques beyond V1 (V4: *Pettine et al., 2019*; V4: *Lu et al., 2018*; area 36: *Takeuchi et al., 2011*). Through histological verification *Takeuchi et al., 2011* found that the prominent current sink with the shortest latency corresponded to the position of the granular layer. We therefore identified this current sink (current sink indicated by white star, between dashed and dotted white lines in *Figure 1F and G*; *Figure 1—figure supplements 1 and 2*) as the granular layer. For each unit, the positions of the electrode contacts (using the five contacts surrounding the one with the largest spike waveform) were weighed by the average peak-to-trough amplitude of the unit's spike waveform on these contacts, to assign the unit to the depth where its spike waveform is largest. By comparing this position with the range of contacts in the granular layer, we could locate units to superficial, granular, or deep layers. Seven penetrations where the CSD map could not be interpreted were excluded from the laminar analyses, and we also required that all units included in the laminar analyses were located ≤2 mm from the most superficial channel with multiunit activity, to minimize the risk of including white matter activity. We evaluated the orthogonality of penetrations by comparing the receptive field (RF) positions for neural activity recorded on different electrode contacts on the probe. To obtain an estimate for these RF positions, for the population of neurons in the vicinity of each electrode contact, we derived multiunit activity as the amplitude of the envelope from the signal on each electrode contact, by band-pass filtering the recorded signal between 500 Hz and 5 kHz, rectifying it and then low-pass filtering it at 200 Hz (adopting the approach from *Self et al., 2019*). RF contours at $z = 3$ were then computed on the $z$-scored root mean square level of this analog signal, using the same procedure as for the spiking data (see *Receptive field mapping*). For each contour, the RF center was defined as the centroid of the region defined by the contour (using MATLAB function *centroid*). A line was then fitted in three dimensions through these RF centers (stacked according to the position of the electrode contact from which they were computed) from a range of electrode contacts covering 2 mm, starting with the most superficial contact that recorded multiunit activity. The variation between RF positions along the probe was then computed as $D$, the distance in the azimuth × elevation plane between the ends of this line, per mm depth ($D$ is indicated for the vertical positions of RF contours shown in *Figure 1H,I* and *Figure 1—figure supplement 1C*, and the population data are described in Results).

## Orientation tuning stimuli

A data set for orientation tuning was obtained using luminance contrast edges similar to those used for RF mapping, but with a circular window, sized and positioned such that the stimulus covered and was centered on the estimated aggregate cRF of the probe (12 orientations; 2 contrast polarities; 54 % luminance contrast; 200 ms stimulus duration).

## Border ownership stimuli and task

From the online analysis on the RF mapping data, position, size, and color of the border ownership stimulus set were chosen. An isoluminant square was positioned on an isoluminant background, as in prior studies on border ownership (*Zhou et al., 2000*). There were four basic conditions (*Figure 1A and C*), consisting of a factorial combination of two square positions and two contrast polarities. Note that this results in two pairs of stimuli, where both scenes in a pair have identical stimulus information inside the cRF (dotted black circles in *Figure 1A and C*). The two luminance areas in each scene were either both grays or a combination of gray and a color, and luminance contrast was 54%, as in prior studies (*Zhou et al., 2000*). Square sizes were between 12 × 12 and 20 × 20 dva. At the beginning

of each trial, a small light gray fixation point (0.2 × 0.2 dva, 80 % luminance contrast) was presented on a blank gray screen (with luminance set at the geometric mean of the luminances in the border ownership stimulus). After the animal maintained fixation for 400 ms, the border ownership stimulus appeared for 500 ms. Then, for a separate project, the stimulus was replaced with a stimulus in which the central edge was prolonged to cover the entire screen (with identical colors and luminances) and the other parts of the square removed, for another 1000 ms. Spikes elicited during that time window were not included in the analyses in this paper. The animal received a juice reward if it maintained fixation throughout the trial. Depending on recording time, data were obtained for different orientations or positions. Conditions were played pseudorandomly in counterbalanced blocks such that each condition was played once before repeating conditions. Typically 8–10 repetitions were obtained per condition. Often these stimuli were played at a few different orientations and/or positions, in order to increase the likelihood of proper stimulus placement for most units recorded on the probe (because the precise cRF for each unit was only available offline, after spike sorting). On some days, the trials analyzed here were randomly interdigitated with similar trials using stimuli defined for other projects. Some of these trials – not analyzed here – included a condition in which the animal had to saccade to a new fixation position if the fixation point moved.

## Analysis

Data were analyzed in MATLAB (MathWorks, Natick, MA). Circular statistics were computed using the MATLAB toolbox CircStat (*Berens, 2009*). Statistical tests for the different analyses are described in detail below. Statistical significance was defined as p < 0.05.

## Spike sorting

The data were sorted offline using SpyKING CIRCUS (*Yger et al., 2018*). The clusters resulting from the automatic sorting step were curated manually using the MATLAB GUI provided by the SpyKING CIRCUS software. Well-isolated units were identified based on a well-defined refractory period in the interspike-interval histogram. Multiunit clusters included in the analysis had to pass a criterion for the signal-to-noise ratio: peak-to-peak amplitude of the average waveform had to exceed five times the standard deviation of the signal 5 ms prior to the peak (similar to *Kashkoush et al., 2019*), after high-pass filtering the data (1-pole butterworth filter, cutoff 300 Hz, implemented using functions *butter* and *filtfilt* in MATLAB).

## Receptive field mapping

To determine the cRF, spikes were counted in a window [30 100] ms after each stimulus onset. The resulting mean counts per stimulus position were transformed to *z*-scores by first subtracting the mean of and then dividing by the standard deviation of spike counts occurring in a window of the same size preceding that stimulus position (*Keliris et al., 2019*). Using the stimulus positions, *z*-scores were transformed to a spatial map, which was smoothed with a Gaussian filter (MATLAB function *imgaussfilt* with $\sigma = 1$). The outline of the cRF was defined as the contour at $z = 3$ on this smoothed map (calculated using MATLAB function *contourc*).

## Border ownership selectivity

Border ownership responses were obtained by recording evoked responses to stimuli as in *Figure 1A and C* (see *Border ownership stimuli* above). A unit's response was evaluated for border ownership selectivity if it passed the following inclusion criteria: (1) average spike rate was ≥1 spike/s for at least one of the four conditions; (2) the evoked spike count for at least one of the four conditions was significantly different from that recorded prior to all trials across conditions (two-sided Wilcoxon rank sum test [MATLAB function *ranksum*] with Bonferroni correction); (3) at least six trials per condition were available; (4) the central edge of the squares in the border ownership stimulus intersected the cRF; (5) the distance between any part of the cRF contour and any part of the noncentral edges of the squares in the border ownership stimulus was ≥1 dva. Data sets obeying these inclusion criteria were candidate data sets for border ownership selectivity. The border ownership index (*BOI*) (*Zhou et al., 2000*) was then calculated for these data sets, which is defined as

$$BOI = \frac{(R_1+R_3)-(R_2+R_4)}{(R_1+R_3)+(R_2+R_4)}$$

where $R_i$ represents the average spike rate in the window [50 500] ms after stimulus onset for condition $i$ (numbering as in **Figure 1A and C**). Statistical significance of border ownership selectivity was evaluated using a permutation test: a null distribution was created by shuffling the border ownership stimulus labels 10,000 times, separately for each luminance contrast pair, and the p value was estimated as the fraction of $|BOI_{shuffled}|$ that was at least as large as $|BOI|$. A data set was defined to be border ownership-selective if p < 0.05 (with Bonferroni correction if multiple orientations or positions were available for the same unit), and a unit was defined to be border ownership-selective if it had at least one border ownership-selective data set.

## Time course of evoked response

For the time course of evoked activity of border ownership-selective units (**Figure 2A–D**), each unit contributed one data set, that is the border ownership-selective data set – as defined above – for which $|BOI|$ was maximal. For each of these, the time courses of the responses to the preferred side of border ownership (side resulting in the highest spike rate) and to the non-preferred side of border ownership (side resulting in the lowest spike rate) were calculated separately (respectively, solid red lines and dashed blue lines **Figure 2A–D**) as follows. Spike trains were rounded to 0.1 ms resolution and convolved with a postsynaptic kernel $K(t)$ (**Thompson et al., 1996**)

$$K\left(t\right) = \left(1 - e^{\frac{-t}{\tau_g}}\right) \cdot e^{\frac{-t}{\tau_d}}$$

where $\tau_g$ = 1 ms and $\tau_d$ = 20 ms. The resulting traces were averaged per condition, and then across both contrast polarities. These average traces were normalized for each unit's evoked response by dividing them by the average value across conditions in the window [50 500] ms after stimulus onset. The mean ± SEM of the resulting functions across units are shown in **Figure 2A–D**. The latency of statistical difference between the functions to the preferred and the non-preferred functions (asterisks in **Figure 2A–D**; **Figure 2—figure supplement 1**) was defined as the first time after which the functions differed statistically (Wilcoxon sign rank test p < 0.05) for 20 adjacent milliseconds.

BOI functions (**Figure 2E**) were defined as the difference between the response function for the preferred side of border ownership and the function for the non-preferred side of border ownership, divided by their sum. Latency of these functions was defined as the earliest crossing of a fixed threshold that was followed by values above the threshold for 20 consecutive milliseconds. The threshold used was derived from shuffled data, by shuffling the stimulus labels for each laminar compartment (1000 shuffles) and finding the lowest value for which <1% of shuffles resulted in a defined latency. Since these values depend on sample size, the highest value across compartments was used as threshold, so that the different functions could be timed using the same threshold. Confidence intervals (95%) were calculated on these latencies using a bootstrap approach with the bias corrected and accelerated percentile method (MATLAB function *bootci*, 2000 bootstraps). The latencies were statistically compared between laminar compartments using a bootstrap approach similar to other studies (**Self et al., 2019**), by computing the difference in latency between bootstrap samples of different compartments and estimating p as the fraction of samples on which the difference was ≤0 (one-sided test).

Spike response functions to small rings in the cRF (**Figure 3**) were computed similarly as the response functions to border ownership stimuli (**Figure 2A–D**). Responses were normalized by dividing them by the peak response for each unit. Latency was defined using a threshold a third of the way from baseline to peak (0.435) using the highest baseline across compartments, and confidence intervals and statistical tests were computed in the same way as for the BOI functions.

## Border ownership reliability and latency

We also evaluated the time course of border ownership selectivity using the metric border ownership reliability (*BOR*; **Zhou et al., 2000**), which reflects the trial-to-trial reliability of encoding border ownership. This metric was computed for the border ownership-selective units in each laminar compartment, in 100 ms sliding windows (1 ms steps). For each unit, 10,000 sets of four spike trains were generated, where each set contained one random spike train from each of the four conditions in **Figure 1A and C**. For each window position, spikes were counted for each spike train in each set. *BOR* for a unit at a particular window position was defined as

$$BOR = \frac{\sum_j A(j)}{\sum_j A(j) + \sum_j B(j)}$$

where $j$ corresponds to the index of all spike train sets for the unit. $A(j)$ and $B(j)$ indicate whether the sign of the spike count difference between border ownership conditions for spike train set $j$ is, respectively, the same or opposite compared to the unit's preferred side of border ownership:

$$\begin{cases} A(j) = 1 \; if \; sgn[(C_{j,1} + C_{j,3}) - (C_{j,2} + C_{j,4})] = S \\ A(j) = 0 \; if \; sgn[(C_{j,1} + C_{j,3}) - (C_{j,2} + C_{j,4})] \neq S \\ B(j) = 1 \; if \; sgn[(C_{j,1} + C_{j,3}) - (C_{j,2} + C_{j,4})] = -S \\ B(j) = 0 \; if \; sgn[(C_{j,1} + C_{j,3}) - (C_{j,2} + C_{j,4})] \neq -S \end{cases}$$
$$S = sgn[(R_1 + R_3) - (R_2 + R_4)]$$

where $C_{j,i}$ represents the window spike count for condition $i$ in spike train set $j$, $R_i$ is the average spike rate for condition $i$ (for the interval [50 500] ms after stimulus onset), and $sgn$ is the sign function. For each window position for each unit, $BOR$ was only computed if there were at least 10 spikes across conditions. For every window position the mean was calculated across units per laminar compartment, resulting in the $BOR$ functions shown in **Figure 2F**. The abscissa in **Figure 2F** corresponds to the position of the right edge of the window, that is it indicates the latest spike times that may have determined $BOR$ for that window position (and thus represents a conservative estimate for the latency). Definition of latency, computation of confidence intervals, and statistical tests used to compare latencies between layers were similar to those for the BOI functions.

## Clustering of preferred side of border ownership

For a given penetration, for each orientation, the border ownership-selective data set with the highest |$BOI$| was selected for each unit. Then from this group of data sets the largest subgroup that shared the same edge orientation and position was retained for analysis. Each unit could thus maximally contribute one data set. The preferred side of border ownership was then determined for all these data sets (example penetrations are shown in **Figure 4C**). The proportion of units sharing the most common preferred side was calculated for each penetration. The average of these proportions across penetrations ($P_{pref}$) was compared with a null distribution generated by randomly assigning the preferred side of border ownership for each data set (2000 randomizations; that is, a binomial process with chance of success = 0.5). The p value was estimated as the fraction of the null distribution for which $P_{pref}$ was at least as large as the actual data.

## Orientation tuning

Orientation selectivity was evaluated using a Kruskal–Wallis test (MATLAB function *kruskalwallis*) on the evoked spike counts in the analysis window (between 30 ms and 200 after stimulus onset) from the orientation tuning data set, using orientation as the grouping variable (**Pettine et al., 2019**), and defined as p < 0.05. All units for which the center of the contrast edge in the orientation data set was positioned in the cRF, and the z-scored spike rate for the orientation with the highest rate was ≥3 were included. Z-scores were calculated by first subtracting the mean of and then dividing by the standard deviation of the spike rate in a window preceding the first stimulus in the trial, equal in duration to the analysis window. For each orientation-selective unit $i$, the response to each orientation $j$ is summarized as $\vec{R}_j$, which has a direction 2 $j$ (because orientation has a period of 180°) and magnitude equal to the spike rate. The resultant $\vec{O}_i$ (vectors shown as solid lines in **Figure 5**) for all $\vec{R}_j$ is then calculated. The magnitude of orientation selectivity and the preferred orientation of unit $i$ were defined, respectively, as the magnitude and as the direction divided by 2 (because of the definition of direction of $\vec{R}_j$) of $\vec{O}_i$.

For each penetration, the aggregate preferred orientation was defined as the direction divided by 2 of the resultant vector $\vec{O}$ of all $\vec{O}_i$ on that penetration (direction of $\vec{O}$ is indicated by blue dashed lines in **Figure 5**). Statistical significance of the aggregate preferred orientation was assessed by randomizing the directions of $\vec{O}_i$ (2000 randomizations), calculating the null distribution $\vec{O}_{shuffled}$, and estimating the p value as the fraction of $\vec{O}_{shuffled}$ for which the magnitude was at least as large as that of $\vec{O}$.

The independence between the fractions of, respectively, orientation-selective and border ownership-selective units (*Table 1*) was assessed using a Chi-square test (MATLAB function *crosstab*).

For units that were border ownership-selective to multiple orientations, the circular mean of border ownership-selective orientations (ordinate in *Figure 6C*) was calculated as the direction divided by 2 of the resultant vector of all vectors $\vec{B}_j$ that have a direction $2j$ and a magnitude equal to $|BOI|$, for all border orientations $j$. The shortest distance of the data points in *Figure 6C* to the identity line (*Figure 6D*) was analyzed by comparing the mean to a null distribution generated by shuffling the values for the preferred edge orientation and the circular mean of border ownership-selective orientations, and calculating the mean of the shortest distance to the identity line (2000 shuffles). The p value was estimated as the fraction of the null distribution that was as small or smaller than the observed value.

## Acknowledgements

This research was supported by NEI core grant for vision research P30-EY019005 to The Salk Institute for Biological Studies. We thank Dr. Edward Callaway, Dr. Anirvan Nandy, and Dr. Zachary Davis for helpful discussions. We thank Dr. Mathias LeBlanc, Dr. Sean Adams, and Ms. Catherine Williams for excellent animal care.

## Additional information

### Funding

| Funder | Grant reference number | Author |
| --- | --- | --- |
| George E. Hewitt Foundation for Medical Research | Postdoctoral Fellowship | Tom P Franken |
| Brain and Behavior Research Foundation | NARSAD Young Investigator Grant | Tom P Franken |
| National Eye Institute | K99EY031795 | Tom P Franken |
| Fiona and Sanjay Jha Chair in Neuroscience | | John H Reynolds |
| Brain and Behavior Research Foundation | 26229 | Tom P Franken |

The funders had no role in study design, data collection and interpretation, or the decision to submit the work for publication.

### Author contributions

Tom P Franken, Conceptualization, Formal analysis, Funding acquisition, Investigation, Software, Visualization, Writing – original draft, Writing – review and editing; John H Reynolds, Conceptualization, Funding acquisition, Supervision, Writing – review and editing

### Author ORCIDs

Tom P Franken http://orcid.org/0000-0001-7160-5152
John H Reynolds http://orcid.org/0000-0001-6988-4607

### Ethics

This study was performed in accordance with the recommendations in the Guide for the Care and Use of Laboratory Animals of the National Institutes of Health. All procedures were approved by the Institutional Animal Care and Use Committee of the Salk Institute for Biological Studies (protocol 14-00014).

### Decision letter and Author response

Decision letter https://doi.org/10.7554/eLife.72573.sa1
Author response https://doi.org/10.7554/eLife.72573.sa2

## Additional files

### Supplementary files
• Transparent reporting form

### Data availability
MATLAB figures with embedded data, and the data plotted as histograms in Figures 4D and 6D are publicly available on figshare (https://doi.org/10.6084/m9.figshare.16862299).

The following dataset was generated:

| Author(s) | Year | Dataset title | Dataset URL | Database and Identifier |
|---|---|---|---|---|
| Franken T, Reynolds J | 2021 | MATLAB figures and data for the article "Columnar processing of border ownership in primate visual cortex" | https://doi.org/10.6084/m9.figshare.16862299 | figshare, 10.6084/m9.figshare.16862299 |

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
