## [Editor Report]

Franken and Reynolds use multielectrode laminar recordings in V4 of macaques to study whether border ownership (BO) occurs with varying latency across layers, whether their ownership preference is consistent across layers, and the relationship between BO and orientation preference. They convincingly show that BO emerges in deep layers and is consistent across depth. Based on these findings, they suggest that BO could arise from horizontal and feedback connections rather than feedforward connections.

---

## [Decision Letter]

**Decision letter after peer review:**

Thank you for submitting your article "Columnar processing of border ownership in primate visual cortex" for consideration by *eLife*. Your article has been reviewed by 3 peer reviewers, one of whom is a member of our Board of Reviewing Editors, and the evaluation has been overseen by Joshua Gold as the Senior Editor. The following individuals involved in review of your submission have agreed to reveal their identity: Rudiger von der Heydt (Reviewer #2); Alexander Thiele (Reviewer #3).

Essential revisions:

1) Determination of layers: The CSD based method used to determine the layers seems a bit ad hoc, although other studies have often used a similar approach. Some histological evidence would be great. If that is not possible, the authors should provide some more details to determine the layer specificity. For example, where were the supragranular-granular and granular-infragranular borders for different penetrations (i.e., which electrode(s) marked these boundaries)? These could be expressed as fractions of the shaft length, and from that, we would approximately know the depth. Also, were these results affected by how the CSDs were smoothed?

2) Another important factor is the orthogonality of the penetrations. This can also be better quantified based on the variation of the RF centers with depth.

3) The authors mentioned the possibility of BO signals being generated in one layer and then relayed to another. Many studies have used directionality measures such as Granger causality to test such predictions. Given the authors have the LFP data, it would be interesting to see this analysis done to test potential inter-layer communication. However, previous studies have studied Granger Causality across, but not within an area, so this analysis may not be suitable for this study. The authors should nonetheless present some results showing the same if that is indeed the case, or at least discuss if this approach is unlikely to work within an area.

4) There is one previous study (Bushnell et al., J Neurosci, 2011) that also concluded, based on the latency of the effect, that border ownership selectivity in V4 cannot be simply inherited from upstream areas. They tested curvature selective V4 cells and found that the responses to the optimal shape were suppressed when the critical curved contour was not intrinsic to the shape but owned by an occluding object. They found latencies of suppression as short as 46 ms, shorter than the latencies of V2 cells reported in Zhou et al., 2000. The present study should mention this finding.

5) These comments would be interesting as discussion points: It would be very interesting to see a similar laminar analysis applied to area V2. Perhaps the computation of border ownership there occurs also in the deep layers, using similar sources of context information. Alternatively, the V2 border ownership selectivity could be the result of back projection from V4. Because of the importance of representing object structure for many visual tasks, as pointed out above, I would not be surprised if similar fast border ownership computation would be generally found in the deep cortical layers, in V1 and V2 and in areas of the ventral stream beyond V4. The new findings support the idea of the grouping cell model, namely that the context information is provided by an external grouping signal that modulates the activity of the feature neurons in the visual cortex. The grouping signal is supposed to represent a 'proto-object', a computational structure that (1) links the visual feature signals, (2) is being remapped across eye movements, and (3) serves object-selective attention. Compared to the large number of studies of feature selectivity in visual cortex, the question of the representation of object structure has prompted relatively few studies, despite its theoretical importance, and the big question of where and how grouping signals are generated still awaits an answer.

6) They speculate that it is computed de novo by infragranular neurons, but there is no proof for that. It could also be is due to feedback from higher areas. Kindly discuss this point.

7) It is hard to see clustering around the identity line in figure 6C.

*Reviewer #1 (Recommendations for the authors):*

The authors mentioned the possibility of BO signals being generated in one layer and then relayed to another. Many studies have used directionality measures such as Granger causality to test such predictions. Given the authors have the LFP data, it would be interesting to see this analysis done to test potential inter-layer communication. However, previous studies have studied Granger Causality across, but not within an area, so this analysis may not be suitable for this study. The authors should nonetheless present some results showing the same if that is indeed the case, or at least discuss if this approach is unlikely to work within an area.

*Reviewer #3 (Recommendations for the authors):*

I find it hard to see clustering around the identity line in figure 6C.

---

## [Author Response]

Essential revisions:1) Determination of layers: The CSD based method used to determine the layers seems a bit ad hoc, although other studies have often used a similar approach. Some histological evidence would be great. If that is not possible, the authors should provide some more details to determine the layer specificity. For example, where were the supragranular-granular and granular-infragranular borders for different penetrations (i.e., which electrode(s) marked these boundaries)? These could be expressed as fractions of the shaft length, and from that, we would approximately know the depth. Also, were these results affected by how the CSDs were smoothed?

We thank the Reviewer for the suggestions. Using CSD from individual penetrations to define the position of laminar compartments is a strategy has been used by several different laboratories, not only in primary visual cortex (e.g. Mitzdorf, 1985; Poort et al., 2016; Bijanzadeh et al., 2018), but also in extrastriate areas, such as other studies in V4 (Nandy et al., 2017; Lu et al., 2018, Pettine et al., 2019; Ferro et al., 2021), and in other cortical areas, for example in medial temporal cortex (Takeuchi et al., 2011). We include a new figure that shows the similarities between the CSD profiles from different studies (Figure 1—figure supplement 3). Figure 1—figure supplement 3A shows the population average of the CSD in our data. The similarity between this panel and the individual examples shown in Figure 1 and Figure 1—figure supplements 1 and 2 further highlights the fairly consistent sink-source patterns observed across individual penetrations in our data. Figure 1—figure supplement 3B shows that this pattern is also consistent with that found in macaque V4 in another laboratory (Pettine et al., 2019). Figure 1—figure supplement 3C shows that this pattern is not specific for V4, but also occurs in other areas, such as the medial temporal cortex (Takeuchi et al., 2011).

Importantly, in the latter study Takeuchi et al. were able to verify histologically, after applying electrolytic marks, that the prominent current sink with short latency (white star in Figure 1—figure supplement 3C) corresponds to the granular layer, thus consistent with CSD analysis in V1 (Mitzdorf, 1985). Thus although we are not able to perform such histological verification in our study (one animal has been euthanized, and the other animal is destined to take part in another project), the very similar sink-source patterns between these different studies (Figure 1—figure supplement 3A-C), including with work from others that has been verified histologically, gives us confidence that we can meaningfully use them to assign electrode contacts to granular, superficial and deep layers, as other laboratories have done (e.g. Lu et al., 2018). In addition to the new Figure 1—figure supplement 3, we added language in the corresponding paragraph in Results to explain this better. We clarified in Results that interpretable CSD maps were found in 81 out of 88 penetrations and that only those were used in the laminar analysis (which was already indicated in Methods in the previous version of the manuscript).

As suggested by the Reviewer, we have added the positions of the electrode contacts to the CSD maps in Figure 1 and Figure 1—figure supplement 2 (labels along ordinate on the right of the panels). The electrode contacts on the probe covered 3.1 mm (32 contacts with 0.1 mm distance between adjacent contacts), thus the full depth of the cortex was covered even though the vertical position of the probe varied between penetrations (also because a layer of granulation tissue develops over time between the artificial dura and the pial surface). Therefore, to aid in estimating the depth of the individual penetrations, we indicate the position of the most superficial contact on which multiunit activity was recorded (solid black triangles in Figure 1; Figure 1—figure supplements 1,2; for all cases where this contact could be identified, i.e. if the most proximal contact on the probe did not show multiunit activity). The average position of this contact is shown on the population CSD map (solid black triangle on Figure 1—figure supplement 3A). The advantage of a method that assigns layers for a penetration based on data from the same penetration, as we have used here, as opposed to a method that assigns compartments based on depth derived from a population average, is that the former helps to avoid errors due to variations in probe position, and due to a variable degree of tissue compression that may occur for different penetrations.

To test whether the results were affected by how the CSDs were smoothed, we performed the layer assignment separately using different methods of smoothing. To ensure that we did not bias the results we blinded ourselves to the original layer assignment when applying each method. We find that the laminar position of >97% of well-isolated units was identical as that obtained when using the standard procedure of smoothing the CSDs. This resulted in robust latency differences of border ownership signals between layers, irrespective of which smoothing method was used. These results are presented in a new supplementary figure (Figure 2—figure supplement 3).

2) Another important factor is the orthogonality of the penetrations. This can also be better quantified based on the variation of the RF centers with depth.

We followed the Reviewer’s suggestion and evaluated orthogonality of the penetrations by computing the distance *D* between receptive field centers along the probe (Methods). We show this metric for the vertical positions of receptive field contours shown for the penetrations in Figure 1H, I and Figure 1—figure supplement 1, and describe the population data in the first paragraph of Results (median (IQR) 0.83 º/mm (1.00 º/mm), for all 81 penetrations that were included in the laminar analyses). This indicates that the variation is small relative to the average diameter of the receptive fields (7.36 º), and that the deviation from orthogonality is limited.

3) The authors mentioned the possibility of BO signals being generated in one layer and then relayed to another. Many studies have used directionality measures such as Granger causality to test such predictions. Given the authors have the LFP data, it would be interesting to see this analysis done to test potential inter-layer communication. However, previous studies have studied Granger Causality across, but not within an area, so this analysis may not be suitable for this study. The authors should nonetheless present some results showing the same if that is indeed the case, or at least discuss if this approach is unlikely to work within an area.

We thank the Reviewer for this thoughtful suggestion. As the Reviewer notes, a number of studies have applied Granger Causality analysis to data recorded across areas. We did, however, find two studies that have applied this approach across laminar compartments within an area (e.g. van Kerkoerle et al., PNAS 2014 https://doi.org/10.1073/pnas.1402773111 ; Ferro et al., PNAS 2021, https://doi.org/10.1073/pnas.2022097118). So, we do not think that the laminar nature of the question itself is a problem.

However, it is unclear that this analysis can be applied to the present dataset. In line with prior studies of border ownership, we defined the border ownership signal as the normalized difference in the firing rate of single neurons, across stimulus conditions that differed in terms of which side of the border was object and which side was ground. Thus, the border ownership signal is defined as a difference in firing rates recorded on separate trials. Granger Causality is typically applied to time series that are recorded simultaneously, with a view to determining the degree to which one time series predicts another. We believe that the brain certainly computes border ownership signals in real time, presumably by comparing the responses of oppositely tuned border-ownership-selective neurons. As shown in the manuscript, the neurons recorded from a single laminar electrode penetration tended to share border ownership preference. Therefore, to answer this question one needs to use a different experimental approach to record simultaneously from subpopulations with opposite border ownership tuning. In theory, one could record laminar signals from columns of oppositely tuned neurons, perhaps by using intrinsic imaging to identify oppositely tuned border ownership columns and using these data to guide penetrations. Such data would allow one to define border ownership signals at the level of single trials, and for the different laminar compartments. This would enable one to study the ongoing flow of BO signals across layers throughout the full response. We thank the Reviewer for this suggestion, which we may take up in future studies. We have added a discussion of these issues to the Discussion.

4) There is one previous study (Bushnell et al., J Neurosci, 2011) that also concluded, based on the latency of the effect, that border ownership selectivity in V4 cannot be simply inherited from upstream areas. They tested curvature selective V4 cells and found that the responses to the optimal shape were suppressed when the critical curved contour was not intrinsic to the shape but owned by an occluding object. They found latencies of suppression as short as 46 ms, shorter than the latencies of V2 cells reported in Zhou et al., 2000. The present study should mention this finding.

We thank the Reviewer for the suggestion. We had not included this work initially because it focuses on shape encoding, comparing isolated shapes with identical shapes that occur incidentally in the presence of an occluder, rather than border ownership. We do agree that their results are consistent with the feedback hypothesis of border ownership, given the short latency of the suppression of incidental shapes (63 ms after stimulus onset in the population, as early as 46 ms for an example neuron). A limitation of interpreting their data as border ownership signals is the different stimulus arrangement compared to our study and prior studies of border ownership. In the Bushnell et al. (2011) study, both the shape and the occluder fell within the classical receptive field. Thus the observed suppression could also be due to differences in the local contour geometry at the T-junctions within the classical receptive field. In any case, we do agree that this suppression of incidental shapes could contribute to border ownership signals in V2 in the presence of occluders, through feedback. We have added a section in the Discussion that discusses this work.

5) These comments would be interesting as discussion points: It would be very interesting to see a similar laminar analysis applied to area V2. Perhaps the computation of border ownership there occurs also in the deep layers, using similar sources of context information. Alternatively, the V2 border ownership selectivity could be the result of back projection from V4. Because of the importance of representing object structure for many visual tasks, as pointed out above, I would not be surprised if similar fast border ownership computation would be generally found in the deep cortical layers, in V1 and V2 and in areas of the ventral stream beyond V4. The new findings support the idea of the grouping cell model, namely that the context information is provided by an external grouping signal that modulates the activity of the feature neurons in the visual cortex. The grouping signal is supposed to represent a 'proto-object', a computational structure that (1) links the visual feature signals, (2) is being remapped across eye movements, and (3) serves object-selective attention. Compared to the large number of studies of feature selectivity in visual cortex, the question of the representation of object structure has prompted relatively few studies, despite its theoretical importance, and the big question of where and how grouping signals are generated still awaits an answer.

We have added comments on potential similar functions of deep layer neurons in different areas to the Discussion. We point out that at least for V2 in the macaque it is challenging to obtain orthogonal recordings with linear multielectrodes, given that at most a few mm of V2 is exposed on the lateral convexity of the brain. We agree that our findings are consistent with the grouping cell model. We have added this more explicitly to the Discussion.

6) They speculate that it is computed de novo by infragranular neurons, but there is no proof for that. It could also be is due to feedback from higher areas. Kindly discuss this point.

We agree with the Reviewers that infragranular neurons may (in full or in part) inherit border ownership signals from higher areas. A requirement for the source of such feedback, whether it contains border ownership signals or grouping signals from which border ownership is computed, is that the latency is short enough to be able to explain the early occurrence of border ownership signals in deep layers in V4. Therefore, as suggested by Zhu et al. (2020), an area in the dorsal stream such as LIP may be a more likely candidate source of such feedback than inferotemporal cortex. We have added this to the Discussion.

7) It is hard to see clustering around the identity line in figure 6C.

We agree with the Reviewers that the bias of data points to cluster around the identity lines in Figure 6C is subtle. The bias is easier to evaluate in Figure 6D, which shows the distribution of the shortest distance of each data point in Figure 6C to the identity lines (the identity line appears three times in panel 6C, because of the periodic nature of both plotted variables). More surprising than finding this bias is that it is this subtle (compare this panel for example with the much stricter relation of orientation preference derived from both datasets in Figure 6—figure supplement 1B). Prior studies have studied border ownership selectivity at the preferred orientation. However, our data show that there is no very strict relation between orientation preference and the orientation of borders for which there is border ownership selectivity (Figure 6C; and the individual examples in Figure 6A, middle and right; Figure 6B). We made changes to this paragraph in Results to explain this better.